Prepared for submission to JHEP

# On the resurgent structure of quantum periods

**Jie Gu**[a] **and Marcos Mariño**[b]

[a]*School of Physics and Shing-Tung Yau Center*
*Southeast University, Nanjing 210096, China*

[b]*Département de Physique Théorique et Section de Mathématiques*
*Université de Genève, Genève, CH-1211 Switzerland*

*E-mail:* eij.ug.phys@gmail.com, Marcos.Marino@unige.ch

ABSTRACT: Quantum periods appear in many contexts, from quantum mechanics to local mirror symmetry. They can be described in terms of topological string free energies and Wilson loops, in the so-called Nekrasov–Shatashvili limit. We consider the trans-series extension of the holomorphic anomaly equations satisfied by these quantities, and we obtain exact multi-instanton solutions for these trans-series. Building on this result, we propose a unified perspective on the resurgent structure of quantum periods. We show for example that the Delabaere–Pham formula, which was originally obtained in quantum mechanical examples, is a generic feature of quantum periods. We illustrate our general results with explicit calculations for the double-well in quantum mechanics, and for the quantum mirror curve of local $\mathbb{P}^2$.

# 1 Introduction

Quantum periods are formal power series in $\hbar$ which appear in the all-orders WKB method applied to one-dimensional quantum systems. In the cases of interest for this paper, theses systems are based on a "quantum curve," i.e. on the quantization of a classical complex curve. The quantum periods are then quantum deformations of classical periods of a meromorphic differential on this curve.

Quantum periods first appeared in the work of Voros on the WKB method in conventional one-dimensional quantum mechanics [1, 2], where the quantum curve is just the Schrödinger operator (the exponentials of quantum periods in quantum mechanics are also called Voros symbols). Quantum periods are important since the spectrum of the Schrödinger operator can be obtained from their Borel resummation, in the form of exact quantization conditions [1, 3, 4]. In addition, Voros studied the properties of quantum periods in the framework of the theory of resurgence of Écalle [5], and he determined in various examples the possible location of their Borel singularities and Stokes discontinuities. The work of Voros on quantum mechanics was further developed in [6–8], where it was found that, in the case of general polynomial potentials, the Stokes discontinuities satisfy a simple and universal formula, sometimes called the Delabaere–Pham formula.

Quantum periods are ubiquitous in all problems involving quantum curves. They re-appeared in [9] as a WKB reinterpretation of results of Nekrasov and Shatashvili (NS) [10] on the quantization of Toda-type integrable systems. Building on [9], the authors of [11] studied the quantum periods associated to the mirror curves of toric Calabi–Yau (CY) manifolds, which define quantum deformations of the periods appearing in local mirror symmetry [12]. In [9, 11], the quantum

curve is typically a functional difference operator or a higher order differential operator. Inspired by the results of [13, 14], it has been argued in [15, 16] that the resurgent structure of quantum periods in these more general settings is related to the counting of BPS states in the corresponding supersymmetric/string theories.

Understanding this resurgent structure in the general case turns out to be a subtle problem. Many of the results of [1, 6, 7] are based on the Voros–Silverstone connection formula for the Schrödinger operator [1, 2, 17] (see e.g. [18, 19] for reviews), and extending this formula to the operators which appear in e.g. the quantization of mirror curves is not straightforward. In this paper we follow a different route, based on the holomorphic anomaly equations (HAE). The HAE were originally defined for the free energies of topological string theory in [20, 21]. It was found in [22, 23] that they can be also applied to the refined topological string free energies, and in particular to their NS limit. The NS free energy is closely related to the quantum periods, as noted in [9, 11], but it does not provide a complete description thereof. For example, if the curve has genus one, there are two independent quantum periods, while the NS free energy only provides one quantity. It has been pointed out recently [24] that the Wilson loop vevs of the supersymmetric gauge theories associated to mirror curves are also governed by a set of HAE, and together with the NS free energy they provide a complete set of quantities to describe the quantum periods. In fact, Wilson loops are the functional inverses of the quantum flat coordinates or quantum mirror maps.

In order to understand the resurgent structure of the quantum periods we need their non-perturbative corrections. Based on the pioneering work of [25, 26], we derive these corrections by considering a trans-series extension of the HAE. In the case of the NS free energy, such an extension was already studied in [27, 28]. In this paper we extend this analysis to Wilson loops and we improve the results of [27] in many ways. By exploiting the operator formalism introduced in [27, 28], we find explicit multi-instanton solutions for the NS free energies and Wilson loops. Based on these considerations, and on the study of examples, we make a proposal for the resurgent structure of the quantum periods. We show in particular that the Delabaere–Pham formula follows essentially from the structure of multi-instantons, together with the behavior of the NS free energies at the conifold point.

This paper is organized as follows. In section 2 we review background material on quantum periods and the HAE for free energies and Wilson loops. Section 3 is the core of the paper. We obtain a general trans-series solution for the HAE which includes the resurgent structures which are found empirically. Based on these formal considerations, we propose a general picture of the resurgent structure of the quantum periods, and we show that the Delabaere–Pham formula follows from well-motivated assumptions. We also discuss mixed instanton-"anti-instanton" solutions to the HAE. In section 4 we analyze two concrete examples to illustrate our general theory. The first one is a quantum mechanical example, namely the symmetric double-well potential of quantum mechanics (some background knowledge for this example is collected in the Appendix). The second example is the quantum mirror curve associated to the local $\mathbb{P}^2$ geometry. We verify our conjectures leading to the Delabaere–Pham formula and we make a first analysis of the resurgent structure in those examples. Finally, in section 5 we summarize our conclusions and propose some directions for future work.

This is a companion paper to [29], where we use similar techniques and ideas to obtain exact multi-instanton amplitudes in conventional topological string theory. Some of the technical tools and results used in this paper are presented in [29] in more detail.

## 2 Quantum periods and the holomorphic anomaly

### 2.1 Quantum periods

Our starting point is a complex curve $\Sigma$ defined by the equation

$$H(x, p) = \xi, \tag{2.1}$$

where $\xi$ is a complex modulus. We will suppose that $x$ and $p$ are canonically conjugate variables, so that $p(x)\mathrm{d}x$ is the Liouville one-form. To obtain the quantum curve, we promote $x$, $p$ to Heisenberg operators satisfying

$$[\mathsf{x}, \mathsf{p}] = \mathrm{i}\hbar. \tag{2.2}$$

By an appropriate quantization scheme, which depends on the case at hand, we can turn $H(x, p)$ into an operator $H(\mathsf{x}, \mathsf{p})$, and consider the eigenvalue problem

$$H(\mathsf{x}, \mathsf{p})|\psi\rangle = \xi|\psi\rangle. \tag{2.3}$$

In the WKB method one uses the following ansatz for the wavefunction $\psi(x)$:

$$\psi(x) = \exp\left(\frac{\mathrm{i}}{\hbar} \int^x Y(x'; \hbar)\mathrm{d}x'\right), \tag{2.4}$$

where the function $Y(x; \hbar)$ is a formal power series in $\hbar$:

$$Y(x; \hbar) = \sum_{n=0}^{\infty} p_n(x)\hbar^n. \tag{2.5}$$

The first term $p_0(x) = p(x)$ is the classical momentum function. We can regard $Y(x; \hbar)\mathrm{d}x$ as a quantum deformation of the Liouville one-form, and integrate it along the one-cycles of $\Sigma$. The resulting objects are formal power series in $\hbar$ and they are called *quantum periods*. We will denote them as

$$\Pi_\gamma(\hbar) = \oint_\gamma Y(x; \hbar)\mathrm{d}x, \qquad \gamma \in H_1(\Sigma). \tag{2.6}$$

If one splits $Y$ into the even component and the odd component,

$$Y(x; \hbar) = p_{\text{even}}(x; \hbar) + p_{\text{odd}}(x; \hbar), \tag{2.7}$$

with

$$p_{\text{even}}(x; \hbar) = \sum_{n=0}^{\infty} p_{2n}(x)\hbar^{2n}, \quad p_{\text{odd}}(x; \hbar) = \sum_{n=0}^{\infty} p_{2n+1}(x)\hbar^{2n+1}, \tag{2.8}$$

one finds in all cases of interest that the odd component is a total derivative, so it does not contribute to the quantum periods. We can then write

$$\Pi_\gamma(\hbar) = \sum_{n\geq 0} \Pi_{\gamma,n}\hbar^{2n}, \qquad \Pi_{\gamma,n} = \oint_\gamma p_{2n}(x)\mathrm{d}x. \tag{2.9}$$

We will call $\Pi_{\gamma,0}$ the *classical periods*. The calculation of the coefficients $\Pi_{\gamma,n}$ at high order can be quite involved, even for simple quantum systems.

Let us make a choice of symplectic basis in $H_1(\Sigma)$, with $A_i \cap B^j = \delta_i^j$, $i, j = 1, \cdots, n$. Then, we can regard the quantum periods associated to the $A$-periods as quantum coordinates,

$$t_i(\hbar) = \Pi_{A_i}(\hbar), \tag{2.10}$$

while the $B$-periods define a quantum prepotential, also called NS free energy, through the equation

$$t_{B^i}(\hbar) = \hbar \frac{\partial \mathcal{F}^{\mathrm{NS}}}{\partial t_i} = \Pi_{B^i}(\hbar). \tag{2.11}$$

In other words, quantum periods define a quantum deformation of special geometry[1]. The NS free energy is a formal power series, defined by the expansion

$$\mathcal{F}^{\mathrm{NS}} = \sum_{n=0}^{\infty} \hbar^{2n-1} \mathcal{F}_n, \tag{2.12}$$

and $\mathcal{F}_0$ is sometimes called the prepotential. We note that the NS free energy depends explicitly on a choice of symplectic basis. This choice is also called a choice of frame. Different frames lead to different functions of different quantum $A$-periods.

**Example 2.1.** *Schrödinger operator.* In the Schrödinger case, the function $H(x, p)$ is of the form

$$H(x, p) = \frac{p^2}{2} + V(x), \tag{2.13}$$

and its standard quantization leads to the Schrödinger operator. The complex modulus $\xi$ is identified with the energy. The eigenvalue equation (2.3) becomes

$$-\hbar^2 \psi''(x) + 2(V(x) - \xi)\psi(x) = 0. \tag{2.14}$$

In this case, as it is well-known (see e.g. [19]), $Y(x; \hbar)$ satisfies the Riccati equation

$$Y^2 - \mathrm{i}\hbar \frac{\mathrm{d}Y}{\mathrm{d}x} = \xi - V(x), \tag{2.15}$$

and one finds that indeed $p_{\mathrm{odd}}(x; \hbar)$ is a total derivative

$$p_{\mathrm{odd}}(x; \hbar) = \frac{\mathrm{i}\hbar}{2} \frac{\mathrm{d}}{\mathrm{d}x} \log p_{\mathrm{even}}(x; \hbar). \tag{2.16}$$

We note that the curves appearing in quantum mechanics can be regarded as Seiberg–Witten curves associated to Argryres–Douglas points [30], therefore many of the tools that are used in gauge/string theories can be also used in the study of quantum mechanics, like for example the HAE [31].

**Example 2.2.** *Quantum mirror curves.* Classical mirror curves of local CY manifolds are of the form

$$W(\mathrm{e}^x, \mathrm{e}^p) = \xi, \tag{2.17}$$

where the l.h.s. is a polynomial in the exponentiated variables. The quantum problem associated to this curve has been considered in [11, 32–35]. As advocated in [35], one can quantize this curve by using Weyl's prescription. The quantum $A$-periods (2.10) are quantum deformations of the flat coordinates defining the mirror map, and we will call them *quantum mirror maps*. By a reasonable abuse of language, we will also use this name to refer to generic quantum $A$-periods.

---

[1]In examples, the quantum coordinates and prepotential are defined by (2.10) and (2.11) up to overall multiplicative factors.

The quantization of Seiberg–Witten curves first considered in [9] is an intermediate example between the Schrödinger operator of conventional quantum mechanics, and the quantization of mirror curves. The quantum problem associated to Seiberg–Witten curves has been studied in many references, see e.g. [16, 36–39].

## 2.2 Free energy, Wilson loops, and the holomorphic anomaly

The NS free energies $\mathcal{F}_n$ appearing in (2.12) can be upgraded to non-holomorphic functions of the moduli of the curve (2.1), as it happens in the closely related case of the conventional topological string free energies [20, 21]. As noted in [22, 23] in the case of local CY manifolds, these functions satisfy holomorphic anomaly equations, similar to those studied in [21]. In the case of the Schrödinger operators, the applicability of the HAE was noted and exemplified in [31]. We will denote the non-holomorphic NS free energies by $F_n$, to distinguish them from their holomorphic limit $\mathcal{F}_n$. We note that $F_1$ are themselves holomorphic, so that $F_1 = \mathcal{F}_1$, and we will use both notations interchangeably. In what follows we will restrict ourselves to the case in which the curve $\Sigma$ is of genus one and has a single "true" modulus (it can have many "mass parameters," see e.g. [40] for the distinction between the two). This complex modulus will be denoted by $z$. In the cases we will consider in this paper, $z$ is related in a simple way to the parameter $\xi$ appearing in the curve (2.1).

The non-holomorphic free energies depend on $z$ and on a propagator function, denoted by $S$, which encodes all the non-holomorphic dependence. For this reason, we will denote them by $F_n(S, z)$. This propagator is the same one appearing in the HAE of the conventional topological string, and its main properties are reviewed in [29], to which we refer for more details. The holomorphic limit of the NS free energies can be obtained by taking the holomorphic limit of the propagator, denoted by $\mathcal{S}$, so that

$$\mathcal{F}_n(z) = F_n(S = \mathcal{S}(z), z). \tag{2.18}$$

The NS free energies in different frames are related by an integral transform [41], but in the context of the HAE they can be simply obtained by making different choices for the holomorphic limit of the propagator. One of the most important properties of the propagator is that its derivative w.r.t. $z$ is a quadratic polynomial in $S$,

$$\partial_z S = S^{(2)}, \qquad S^{(2)} = C_z \left( S^2 + 2\mathfrak{s}S + \mathfrak{f} \right). \tag{2.19}$$

Here, $C_z$ is the Yukawa coupling, which is defined by

$$\partial_t^3 F_0 = C_t = \left( \frac{\mathrm{d}z}{\mathrm{d}t} \right)^3 C_z. \tag{2.20}$$

The HAE equations determine the dependence of $F_n(S, z)$ on the propagator, once the lower order free energies $F_{n'}(S, z)$, $n' < n$, are known. In our approach to the HAE, we regard $S$ as an independent argument, and we trade the derivatives w.r.t. $z$ of $S$ by a quadratic polynomial in $S$, by using (2.19). To do this, it is useful to introduce a derivation $\mathfrak{D}_z$, which acts as

$$\mathfrak{D}_z f(S, z) = \partial_S f(S, z) S^{(2)} + \partial_z f(S, z), \tag{2.21}$$

on arbitrary functions of $S$ and $z$. In the holomorphic limit, $\mathfrak{D}_z$ becomes the usual derivative w.r.t. $\partial_z$.

The starting point of the HAE is $F_1$, which is a holomorphic function. We will write it as

$$F_1 = \frac{1}{24} \log f_1(z). \tag{2.22}$$

The holomorphic anomaly equations govern the NS free energies for $n \geq 2$. In the one-modulus case they can be written as

$$\frac{\partial F_n}{\partial S} = \frac{1}{2} \sum_{m=1}^{n-1} \mathfrak{D}_z F_m \mathfrak{D}_z F_{n-m}. \tag{2.23}$$

It is straightforward to obtain $F_2$ from this equation:

$$F_2 = \frac{1}{2} \left( \mathfrak{D}_z F_1 \right)^2 S + f_2(z) = \frac{1}{1152} \left( \frac{f_1'(z)}{f_1(z)} \right)^2 S + f_2(z), \tag{2.24}$$

where $f_2(z)$ is an arbitrary holomorphic function of $z$ which is not fixed by the HAE. This happens at all orders $n \geq 2$, and the resulting functions $f_n(z)$ are called *holomorphic ambiguities*. In order to fix these, we need additional conditions, not included in the HAE. Such conditions can be obtained from the behavior of the NS free energies at special points in moduli space. To understand these conditions, we have to be more specific about the way in which the HAE encode the dependence on the moduli. Since we are in the one-modulus case, there will be a quantum $A$-period, which we will denote by $t$, and therefore a single quantum mirror map, which can be written as

$$t(z; \hbar) = \Pi_A(z; \hbar) = \sum_{n \geq 0} t_n(z) \hbar^{2n}. \tag{2.25}$$

The classical limit of the period $t_0(z)$ defines the classical mirror map, which we will simply denote by $t(z)$. For our purposes it is crucial to note that the modulus $z$ appearing in the solution of the HAE is related to the quantum $A$-period by the *classical* mirror map [23], and it could be regarded as a "quantum" modulus. However, to make notations simpler, we will denote it by $z$. Therefore, given the holomorphic free energies $\mathcal{F}_n(z)$ obtained from the HAE, we can restore the full dependence on $t(z; \hbar)$ by simply using the *classical* inverse mirror map $z = z(t)$.

Let us now come back to the issue of boundary conditions. There are conifold points at the moduli space of the curve where the NS free energies in the appropriate frame behave as [22, 23]

$$F_n = \mathfrak{a} \mathfrak{b}^{n-1} \frac{(1 - 2^{1-2n})(2n-3)!}{(2n)!} \frac{B_{2n}}{t_c^{2n-2}} + \mathcal{O}(1), \qquad n \geq 2, \tag{2.26}$$

where $B_{2n}$ are Bernoulli numbers, $\mathfrak{a}$, $\mathfrak{b}$ are constants which depend on the particular model, and $t_c$ is the appropriate coordinate near the conifold point. In the case of Schrödinger operators for potential wells, for example, conifold points in moduli space correspond to values of the energies at the bottom or the top of the potential well. The behavior (2.26) follows already from the considerations of [6, 7] and was explored in detail in [31].

Finally, it will be useful to rewrite the HAE as a "master equation" governing the full perturbative series. By using the modified free energy

$$\widetilde{F} = \hbar F^{\mathrm{NS}} - F_0 = \sum_{n \geq 1} F_n \hbar^{2n}, \tag{2.27}$$

one finds [27]

$$\frac{\partial \widetilde{F}}{\partial S} = \frac{1}{2} \left( \mathfrak{D}_z \widetilde{F} \right)^2. \tag{2.28}$$

Let us now turn to Wilson loops (more precisely, Wilson loop vevs). Originally, these quantities are defined for local CY geometries which engineer five and four dimensional gauge theories [42], and we refer to [24] and e.g. [37] for background and references. We are only interested here in Wilson loops in the NS limit, and we will restrict ourselves to the one-modulus case. Wilson loops are given by asymptotic expansions of the form

$$\omega(z; \hbar) = \sum_{n \geq 0} \omega_n(z) \hbar^{2n}, \tag{2.29}$$

where

$$\omega_0(z) = c \log(z) \tag{2.30}$$

and $c$ is a constant that depends on the geometry. In order to obtain the quantum periods, the value of $c$ is not important, so from now on we will set $c = 1$. As we will review in more detail in a moment, the Wilson loops $\omega_n$ satisfy HAE, just as the NS free energy. Also as in that case, their natural argument $z$ is the "quantum" version of the modulus. This means that to obtain their dependence on the full *quantum A*-period we also use the *classical* inverse mirror map $z = z(t)$, as in the case of the NS free energies.

From the point of view of this paper, the most important property of the Wilson loop is that it is the *inverse* of the quantum mirror map or quantum $A$-period. More precisely, we have

$$\omega\left(z\left(t(z; \hbar)\right); \hbar\right) = \log(z). \tag{2.31}$$

In using this equation we have to be careful. The first occurrence of $z$ in (2.31), in the argument of $\omega$ in the l.h.s., is the "quantum" version of the modulus. We recall that this is a function of the quantum mirror map $t(z; \hbar)$, given by the classical inverse mirror map. The second occurrence of $z$, in the argument of $t$ and in the r.h.s., is the ordinary modulus of the curve. As an illustration, let us use (2.31) to obtain the first quantum correction to the mirror map in (2.25) from the quantum corrections to the Wilson loop in (2.29). We have

$$z(t(z; \hbar)) = z + \frac{\mathrm{d}z}{\mathrm{d}t} t_1(z) \hbar^2 + \left\{ \frac{\mathrm{d}z}{\mathrm{d}t} t_2 + \frac{1}{2} \frac{\mathrm{d}^2 z}{\mathrm{d}t^2} (t_1(z))^2 \right\} \hbar^4 + \mathcal{O}(\hbar^6),$$

$$\omega_n\left(z(t(z; \hbar))\right) = \omega_n(z) + \frac{\mathrm{d}\omega_n}{\mathrm{d}z} \frac{\mathrm{d}z}{\mathrm{d}t} t_1(z) \hbar^2 + \cdots \tag{2.32}$$

Therefore, (2.31) gives for the first order

$$t_1(z) = -z \frac{\mathrm{d}t}{\mathrm{d}z} \omega_1(z). \tag{2.33}$$

In [24], a HAE is proposed for Wilson loops associated to local CY manifolds, i.e. to curves of the form (2.17). This equation governs the non-holomorphic version of the Wilson loops, which we will denote by $w_n(S, z)$. The HAE reads,

$$\frac{\partial w_n}{\partial S} = \sum_{k=0}^{n-1} \mathfrak{D}_z w_k \mathfrak{D}_z F_{n-k}, \tag{2.34}$$

where $n \geq 1$ and $F_r$ are the perturbative NS free energies. This can be integrated systematically, order by order in $n$, and one finds e.g. for the first term,

$$w_1(S, z) = \frac{1}{24z} \frac{f_1'(z)}{f_1(z)} S + w_1^h(z), \tag{2.35}$$

where $f_1(z)$ is the holomorphic function in (2.22), and $w_1^h(z)$ is the holomorphic ambiguity. As in the case of the free energies, this ambiguity appears at all orders $n \geq 1$. It can be fixed (at least partially) by requiring the appropriate behavior at special points. It is argued in [24] that, in the case of toric CYs, the holomorphic limit $\omega_n(z)$ in the conifold frame has to be regular at the conifold point. The HAE (2.34) were studied in detail in [24] for various local geometries, like local $\mathbb{P}^2$ or local $\mathbb{F}_0$, and it was checked explicitly that regularity at the conifold is enough to fix the holomorphic ambiguities. The HAE (2.34) should be valid as well for Schrödinger operators, and we have verified explicitly that this is the case for the double well potential in quantum mechanics [27, 31]. In Appendix A we give some details on the HAE in that case.

As in the case of the free energy, we will need a master equation for the full formal sum,

$$w(S, z; \hbar) = \sum_{n \geq 0} w_n(S, z)\hbar^{2n}. \tag{2.36}$$

It is easy to see that the master equation is given by

$$\frac{\partial w}{\partial S} = \mathfrak{D}_z w \, \mathfrak{D}_z \widetilde{F}, \tag{2.37}$$

where $\widetilde{F}$ is defined in (2.27) (we have taken into account that $w_0 = \omega_0$ does not depend of $S$).

The NS free energy and the Wilson loops are enough to determine both the quantum $A$-period and the quantum $B$-period (remember that we are restricting ourselves to the case in which $\Sigma$ has genus one): given a choice of frame, the quantum $A$-period is obtained by inverting the Wilson loop, as in (2.31), while the quantum $B$-period is given by

$$t_B(z; \hbar) = \hbar \frac{\partial}{\partial t} F^{\mathrm{NS}}(z(t(z; \hbar)); \hbar). \tag{2.38}$$

Let us note that the $B$-period defines a dual frame. Therefore, another way to obtain the same information is to consider the Wilson loop and (2.31) in that dual frame:

$$\omega_B\left(z\left(t_B(z; \hbar)\right); \hbar\right) = \log(z). \tag{2.39}$$

These two perspectives give complemenary information on the quantum $B$-periods.

## 3 Trans-series solutions and resurgent quantum periods

### 3.1 Trans-series extension of the holomorphic anomaly equations

In the previous section we have introduced two formal power series in $\hbar^2$, the NS free energies and the Wilson loop, from which we can deduce the two quantum periods associated to a genus one quantum curve. We would like to know what are the "instanton sectors" associated to these series. These are exponentially small corrections in $\hbar$, and we expect them to appear as trans-series associated to the Borel singularities of $F^{\mathrm{NS}}$ and $w$. This is a first step in determining the so-called resurgent structure associated to these two series (we will recall below, following e.g. [29, 43], what are the basic ingredients of a resurgent structure). Our strategy to obtain these instanton sectors or formal trans-series was first put forward in [25, 26] for the conventional topological string, and extended in [27, 28] to the NS free energy: one considers the HAE but use a trans-series ansatz for its solution, including exponentially small corrections.

Let us first consider trans-series solutions for the equations governing the NS free energies. Some of these results were already obtained in [27]. We assume that the master equation (2.28) is satisfied with a trans-series ansatz of the form[2],

$$\widetilde{F} = \widetilde{F}^{(0)} + F, \qquad F = \sum_{\ell \geq 1} \mathcal{C}^\ell F^{(\ell)}, \tag{3.1}$$

where $\widetilde{F}^{(0)}$ corresponds to the perturbative sector (2.12) and $\mathcal{C}$ is a trans-series parameter which keeps track of the instanton order. The formal series $F^{(\ell)}$ are interpreted as $\ell$-instanton amplitudes. We will assume that they are of the form

$$F^{(\ell)} = \mathrm{e}^{-\ell \mathcal{A}/\hbar} \sum_{k \geq 0} \hbar^k F_k^{(\ell)}. \tag{3.2}$$

Here, $\mathcal{A}$ is the instanton action. General considerations suggest that $\mathcal{A}$ must be a combination of classical periods of the curve [25, 26, 44], and we will write it as

$$\mathcal{A} = \alpha \partial_t \mathcal{F}_0(t) + \beta t + \gamma. \tag{3.3}$$

We note that $\mathcal{A}$ is a classical period and therefore it defines a frame. The holomorphic propagator associated to this frame will be denoted by $\mathcal{S}_\mathcal{A}$. Let us point out that (3.1) is not the most general ansatz that can be considered, since one should incorporate as well "anti-instantons". We will do this later on.

It is instructive to work out the very first orders in $\hbar$ of, say, the first instanton correction $F^{(1)}$, as it was done in [27]. By plugging the ansatz (3.2) in the master equation (2.28), we obtain

$$\frac{\partial F^{(1)}}{\partial S} = \mathfrak{D}_z F^{(1)} \mathfrak{D}_z \widetilde{F}^{(0)}. \tag{3.4}$$

The first consequence of these equations is that $\partial_S \mathcal{A} = 0$, as in [25, 26]. In terms of components, we find the recursive equations

$$\partial_S F_r^{(1)} = \sum_{k+2n-1=r,\, n \geq 1} \mathfrak{D}_z F_n^{(0)} F_k^{(1)} \mathfrak{D}_z \mathcal{A} + \sum_{k+2n=r,\, n \geq 1} \mathfrak{D}_z F_n^{(0)} \mathfrak{D}_z F_k^{(1)}. \tag{3.5}$$

In detail,

$$\begin{aligned}
\partial_S F_0^{(1)} &= 0, \\
\partial_S F_1^{(1)} &= -\mathfrak{D}_z F_1^{(0)} F_0^{(1)} \mathfrak{D}_z \mathcal{A}, \\
\partial_S F_2^{(1)} &= -\mathfrak{D}_z F_1^{(0)} F_1^{(1)} \mathfrak{D}_z \mathcal{A} + \mathfrak{D}_z F_1^{(0)} \mathfrak{D}_z F_0^{(1)},
\end{aligned} \tag{3.6}$$

and so on. As explained in [27], the first equation says that $F_0^{(1)}$ is a global holomorphic function:

$$F_0^{(1)} = f_0^{(1)}(z). \tag{3.7}$$

This is the non-perturbative counterpart of the holomorphic ambiguity. We can now integrate the second equation in (3.6) to obtain

$$F_1^{(1)} = f_1^{(1)}(z) - f_0^{(1)}(z) \mathfrak{D}_z F_1^{(0)} \mathfrak{D}_z \mathcal{A}\, S, \tag{3.8}$$

---

[2]Note that, with this notation, $F$ involves only non-perturbative sectors. Also note that it has an additional power of $\hbar$ as in (2.27).

where $f_1^{(1)}(z)$ is a new holomorphic ambiguity. It is clear that, in order to make progress, we have to fix the ambiguities arising at the non-perturbative level. There is an ingenuous solution to do this which was found in [25, 26].

One can evaluate the holomorphic limit of the $\ell$-th instanton solution in an arbitrary frame by simply setting $S$ to the appropriate value. It was pointed out in [25–27, 45] that multi-instanton amplitudes associated to an instanton action $\mathcal{A}$ simplify in the frame defined by $\mathcal{A}$ itself, i.e. when $S = \mathcal{S}_{\mathcal{A}}$. Let us consider for example the one-instanton case, and let us denote by $\mathcal{F}_{\mathcal{A}}^{(1)}$ the holomorphic limit in that frame. Then, one has

$$\mathcal{F}_{\mathcal{A}}^{(1)} = \mathrm{e}^{-\mathcal{A}/\hbar}, \tag{3.9}$$

up to an overall multiplicative constant which can be absorbed in the trans-series parameter. The behavior (3.9) can be justified when $\mathcal{A}$ is proportional to the conifold coordinate $t_c$ and we work in the conifold frame, by using the behavior (2.26). If $t_c$ is sufficiently small, the large order behavior of $\mathcal{F}_n$ will be dominated by the pole of order $2n - 2$ in (2.26). The well-known formula for the Bernoulli numbers,

$$B_{2n} = (-1)^{n-1} \frac{2(2n)!}{(2\pi)^{2n}} \sum_{\ell \geq 1} \ell^{-2n}, \tag{3.10}$$

gives the following all-orders asymptotic behavior [28]:

$$\mathcal{F}_n \sim \frac{\mathfrak{a}}{2\pi^2} \Gamma(2n-2) \sum_{\ell \geq 1} \frac{(-1)^{\ell-1}}{\ell^2} (\ell \mathcal{A}_c)^{2-2n}, \qquad n \gg 1, \tag{3.11}$$

where

$$\mathcal{A}_c = \frac{2\pi \mathrm{i}}{\sqrt{\mathfrak{b}}} t_c. \tag{3.12}$$

By using the standard correspondence between large order behavior and exponentially small corrections (see e.g. [46]) we find that (3.11) corresponds to an $\ell$-th instanton amplitude of the form,

$$\frac{(-1)^{\ell-1}}{\ell^2} \mathrm{e}^{-\ell \mathcal{A}_c/\hbar}, \tag{3.13}$$

up to overall factors which do not depend on $\ell$. This suggests the following generalization of the boundary condition (3.9) to the $\ell$-instanton case,

$$\mathcal{F}_{\mathcal{A}}^{(\ell)} = \frac{(-1)^{\ell-1}}{\ell^2} \mathrm{e}^{-\ell \mathcal{A}/\hbar}. \tag{3.14}$$

We can now come back to the calculation of $F^{(1)}$. Since the holomorphic ambiguities do not depend on the frame, we can evaluate them in the frame associated to $\mathcal{A}$. By imposing the boundary condition (3.9) we can easily fix their value and we conclude that

$$F_0^{(1)} = 1, \qquad F_1^{(1)} = -\mathfrak{D}_z F_1^{(0)} \mathfrak{D}_z \mathcal{A}(S - \mathcal{S}_{\mathcal{A}}). \tag{3.15}$$

One finds in addition,

$$F_2^{(1)} = \frac{1}{2} (\mathfrak{D}_z F_1^{(0)})^2 \left[ \mathfrak{D}_z \mathcal{A}(S - \mathcal{S}_{\mathcal{A}}) \right]^2. \tag{3.16}$$

It was noted in [27] in examples that one can solve $F_n^{(1)}$ at all orders, as we will explain in the next section.

Let us now consider the trans-series solution for the Wilson loop. We consider an ansatz

$$w = \sum_{\ell \geq 0} \mathcal{C}^\ell w^{(\ell)}(\hbar), \tag{3.17}$$

where $w^{(\ell)}(\hbar)$ is of the form

$$w^{(\ell)}(\hbar) = \mathrm{e}^{-\ell \mathcal{A}/\hbar} \sum_{n \geq 0} w_n^{(\ell)} \hbar^n. \tag{3.18}$$

As an example, let us solve for $w^{(1)}$ at the very first orders in $\hbar$. By plugging (3.17) in the master equation (2.37), we obtain

$$\frac{\partial w^{(1)}}{\partial S} = \mathfrak{D}_z w^{(0)} \mathfrak{D}_z F^{(1)} + \mathfrak{D}_z w^{(1)} \mathfrak{D}_z \widetilde{F}^{(0)}. \tag{3.19}$$

The first two terms in $w^{(1)}$ satisfy

$$\begin{aligned} \frac{\partial w_0^{(1)}}{\partial S} &= -\mathfrak{D}_z w_0^{(0)} \mathfrak{D}_z \mathcal{A} F_0^{(1)}, \\ \frac{\partial w_1^{(1)}}{\partial S} &= -\mathfrak{D}_z w_0^{(0)} \mathfrak{D}_z \mathcal{A} F_1^{(1)} - w_0^{(1)} \mathfrak{D}_z \mathcal{A} \mathfrak{D}_z F_1^{(0)}. \end{aligned} \tag{3.20}$$

As in the case of the NS free energies, we need additional conditions to integrate these equations. Since the Wilson loop in the conifold frame is smooth at the conifold point, we will assume the boundary condition

$$w_\mathcal{A}^{(\ell)} = 0. \tag{3.21}$$

If we use this boundary condition, together with the values for the one-instanton free energy obtained above, we find

$$w_0^{(1)} = -\mathfrak{D}_z w_0^{(0)} \mathfrak{D}_z \mathcal{A}(S - S_\mathcal{A}), \qquad w_1^{(1)} = \mathfrak{D}_z w_0^{(0)} \mathfrak{D}_z F_1^{(0)} \left(\mathfrak{D}_z \mathcal{A}(S - S_\mathcal{A})\right)^2. \tag{3.22}$$

In the next section we will use the methods of [27] to obtain all-orders solutions for $w^{(\ell)}$.

### 3.2 Multi-instanton solutions

In the previous section we have shown how to solve the HAE for the multi-instantons, order by order in $\hbar$ and in the instanton number, as in [25, 26]. It was noted in [27, 28] in the case of the NS free energy that one can do better and obtain exact solutions for the multi-instanton amplitudes at all orders in $\hbar$. We will now extend and improve the results of [27, 28] to find exact, explicit results for $F^{(\ell)}$ and $w^{(\ell)}$, at all orders in $\hbar$ and for all instanton numbers, for any curve with one modulus (explicit expressions for $F^{(\ell)}$, at all orders in $\hbar$ but only for the very first values of $\ell$, were obtained in [27]). The solutions we will obtain depend on a general choice of boundary conditions and include in particular the trans-series found in actual examples, as we will verify in later sections.

The key idea to obtain these exact solutions is to reformulate the HAE in terms of two operators introduced in [27, 28]. Let us denote

$$T = \mathfrak{D}_z \mathcal{A} (S - S_\mathcal{A}). \tag{3.23}$$

Then, our first operator is

$$\mathsf{D} = T \mathfrak{D}_z. \tag{3.24}$$

The second operator is

$$\mathsf{W} = T^2 \partial_S - \mathsf{D}\widetilde{F}^{(0)}\mathsf{D}. \tag{3.25}$$

We note that $\mathsf{D}$, $\mathcal{D}_S$ and $\mathsf{W}$ are all derivations. It is important to understand the meaning of the operator $\mathsf{D}$ in the holomorphic limit. Let us suppose that the action $\mathcal{A}$ is given by (3.3), and let $\mathcal{S}$ be the holomorphic limit of the propagator in the frame whose natural flat coordinate is $t$. Then, one has [29]

$$\mathcal{S} - \mathcal{S}_\mathcal{A} = \alpha \left( \frac{\mathrm{d}t}{\mathrm{d}z} \frac{\mathrm{d}\mathcal{A}}{\mathrm{d}z} \right)^{-1}, \tag{3.26}$$

and

$$\mathsf{D} \rightarrow \alpha \partial_t. \tag{3.27}$$

We also note that, in the frame defined by $\mathcal{A}$, $T = 0$, and $\mathsf{D}$ acts as zero. Sometimes we will use $\mathsf{D}$ in the holomorphic limit as a compact way of writing $\alpha \partial_t$.

Another object which will play an important rôle in what follows is

$$\mathcal{G} = \mathcal{A} + \mathsf{D}\widetilde{F}^{(0)}. \tag{3.28}$$

To understand this object, we first note that, if $\alpha \neq 0$ in (3.3), $\mathcal{A}$ defines a modified prepotential $\mathcal{F}_0^\mathcal{A}$ through the equation

$$\mathcal{A} = \alpha \partial_t \mathcal{F}_0^\mathcal{A}. \tag{3.29}$$

Then, in the holomorphic limit, we have

$$\mathcal{G} \rightarrow \alpha \hbar \, \partial_t \mathcal{F}^{\mathrm{NS}}, \tag{3.30}$$

where the prepotential appearing in $\mathcal{F}^{\mathrm{NS}}$ is actually $\mathcal{F}_0^\mathcal{A}$. Therefore, when $\alpha \neq 0$, the holomorphic limit of $\mathcal{G}$ is the quantum $B$-period (2.38) in which we have redefined $\mathcal{F}_0$, i.e. it is the quantum period whose classical limit is $\mathcal{A}$. In addition, in the frame defined by $\mathcal{A}$, we have

$$\mathcal{G}_\mathcal{A} = \mathcal{A}. \tag{3.31}$$

The basic ingredient of this operator formulation is the commutation relation,

$$[\mathsf{W}, \mathsf{D}] = \mathsf{D}\mathcal{G}\,\mathsf{D}, \tag{3.32}$$

which was already introduced in [27] in examples. A proof of this relation for general curves with one modulus can be found in [29].

We can now use these operators to rewrite the HAE (2.28), (2.37) for the NS free energy and Wilson loops. For the perturbative free energy, we find

$$\mathsf{W}\widetilde{F}^{(0)} = -\frac{1}{2}\left(\mathsf{D}\widetilde{F}^{(0)}\right)^2, \tag{3.33}$$

while the non-perturbative free energy $F$ introduced in (3.1) satisfies

$$\mathsf{W}F = \frac{1}{2}\left(\mathsf{D}F\right)^2. \tag{3.34}$$

Before proceeding further, let us note that

$$\mathsf{W}\mathcal{G} = 0. \tag{3.35}$$

This can be proved by direct calculation, by using (3.32) and (3.33):

$$
\begin{aligned}
\mathsf{W}\mathcal{G} &= \mathsf{W}\mathcal{A} + \mathsf{W}\mathsf{D}\widetilde{F}^{(0)} = \mathsf{W}\mathcal{A} + \mathsf{D}\mathsf{W}\widetilde{F}^{(0)} + \mathsf{D}\mathcal{G}\mathsf{D}\widetilde{F}^{(0)} \\
&= \mathsf{W}\mathcal{A} + \mathsf{D}\mathcal{G}\mathsf{D}\widetilde{F}^{(0)} - \mathsf{D}^2\widetilde{F}^{(0)}\mathsf{D}\widetilde{F}^{(0)} = \mathsf{W}\mathcal{A} + \mathsf{D}\mathcal{A}\mathsf{D}\widetilde{F}^{(0)} = 0.
\end{aligned}
\tag{3.36}
$$

We can now solve for $F^{(1)}$ in closed form [27]. It satisfies

$$
\mathsf{W}F^{(1)} = 0,
\tag{3.37}
$$

and due to (3.35) it follows that

$$
F^{(1)} = \mathrm{e}^{-\mathcal{G}/\hbar}
\tag{3.38}
$$

is a solution satisfying the boundary condition (3.9). This reproduces the very first orders in (3.15), (3.16).

Let us now consider the HAE (2.37) for the Wilson loop. We have, for the perturbative part,

$$
\mathsf{W}w^{(0)} = 0.
\tag{3.39}
$$

On the other hand, the trans-series version (3.17) satisfies

$$
\mathsf{W}w = \mathsf{D}w\,\mathsf{D}F,
\tag{3.40}
$$

where we recall that $F$, defined in (3.1), only involves instanton sectors. For example, the one-instanton correction to the Wilson loop satisfies

$$
\mathsf{W}w^{(1)} = \mathsf{D}w^{(0)}\mathsf{D}F^{(1)}.
\tag{3.41}
$$

Taking into account the solution for $F^{(1)}$ (3.38), one finds that (3.41) is solved by

$$
w^{(1)} = -\frac{1}{\hbar}\mathsf{D}w^{(0)}\mathrm{e}^{-\mathcal{G}/\hbar},
\tag{3.42}
$$

which indeed satisfies the boundary condition (3.21). One verifies that the very first orders obtained in (3.22) are reproduced by the all-orders solution (3.42), as it should.

It is possible to proceed in this way, order by order in the instanton sector, and obtain solutions for $F^{(\ell)}$, $w^{(\ell)}$. We would like however to obtain expressions in closed form for *arbitrary* values of $\ell$. To do this, we will assume that the boundary conditions are of the form

$$
F_{\mathcal{A}}^{(\ell)} = \tau_\ell \frac{(-1)^{\ell-1}}{\ell^2}\mathrm{e}^{-\ell\mathcal{A}/\hbar},
\tag{3.43}
$$

where $\tau_\ell$ are arbitrary constants. These boundary conditions are suggested by (3.14). The corresponding solutions are general enough to include the trans-series appearing in the resurgent structure, as we will see in the next sections.

Let us construct the solution to (3.34) with the above boundary conditions. We first define

$$
R(y;\mathcal{C}) = -\frac{1}{\hbar}\sum_{j\geq 1}\frac{(-1)^{j-1}}{j}\mathcal{C}^j\tau_j\exp\left\{-\frac{j}{\hbar}\mathrm{e}^{y\mathsf{D}}\mathcal{G}\right\}.
\tag{3.44}
$$

This is a formal power series in $y$ whose coefficients are formal power series in $\mathcal{C}$. We now define $\Sigma(\mathcal{C})$ as the solution of the implicit equation

$$
\Sigma(\mathcal{C}) = R\left(\Sigma(\mathcal{C});\mathcal{C}\right).
\tag{3.45}
$$

$\Sigma(\mathcal{C})$ is a formal power series in $\mathcal{C}$,

$$\Sigma(\mathcal{C}) = \sum_{\ell \geq 1} \mathcal{C}^\ell \Sigma_\ell. \tag{3.46}$$

We have for example

$$\Sigma_1 = -\frac{\tau_1}{\hbar} e^{-\mathcal{G}/\hbar}, \qquad \Sigma_2 = \left(\frac{\tau_2}{2\hbar} - \frac{\tau_1^2}{\hbar^3} \mathsf{D}\mathcal{G}\right) e^{-2\mathcal{G}/\hbar}. \tag{3.47}$$

An explicit expression for $\Sigma(\mathcal{C})$ and the $\Sigma_\ell$ can be obtained by using the Lagrange inversion theorem. Statements of this theorem in forms suitable to our purposes can be found in [47–49]. In particular, we can use Theorems 2.4.1 and 2.4.2 of [47]. In our context, the theorem says that if $H(y;\mathcal{C})$ is a formal power series, and $\Sigma(\mathcal{C})$ is defined by the implicit equation (3.45), then

$$H(\Sigma(\mathcal{C});\mathcal{C}) = H(0;\mathcal{C}) + \sum_{m \geq 1} \frac{1}{m} [y^{m-1}] \frac{\partial H}{\partial y} R^m(y;\mathcal{C}), \tag{3.48}$$

Here, $[y^n]f(y)$ means the term of order $y^n$ in the formal power series $f(y)$. We will also need the equivalent result that

$$\frac{H(\Sigma(\mathcal{C});\mathcal{C})}{1 - \partial_y R(\Sigma(\mathcal{C});\mathcal{C})} = \sum_{m \geq 1} [y^m] H(y;\mathcal{C}) R^m(y;\mathcal{C}). \tag{3.49}$$

Then, we find

$$\Sigma(\mathcal{C}) = \sum_{m \geq 1} \frac{1}{m} [y^{m-1}] R^m(y;\mathcal{C}). \tag{3.50}$$

We now claim that

$$F = -\hbar \int_0^{\mathcal{C}} \Sigma(\mathcal{C}') \frac{\mathrm{d}\mathcal{C}'}{\mathcal{C}'} \tag{3.51}$$

solves (3.34) with the initial conditions (3.43). We first prove (3.34). To this end, we make the following change of variables in the integral appearing in the r.h.s. of (3.51):

$$R(y;\mathcal{C}) = z, \tag{3.52}$$

which defines $z$ as a function of $\mathcal{C}$, parametrized by $y$. Conversely, it defines a function $\mathcal{C}(z;y)$. We then have,

$$\mathrm{d}z = \partial_{\mathcal{C}} R(y;\mathcal{C}) \mathrm{d}\mathcal{C}, \tag{3.53}$$

and we write

$$F = -\sum_{m \geq 1} \frac{\hbar}{m} [y^{m-1}] \int_0^{R(y;\mathcal{C})} \frac{z^m}{r(z)} \mathrm{d}z, \tag{3.54}$$

where

$$r(z) = \mathcal{C} \partial_{\mathcal{C}} R(y;\mathcal{C}). \tag{3.55}$$

It is important to note that $r(z)$ does not depend explicitly on $y$. This can be checked by noticing that $R(y;\mathcal{C})$ is of the form

$$R(y;\mathcal{C}) = \mathfrak{r}\left(\mathcal{C} \exp\left(-e^{y\mathsf{D}}\mathcal{G}/\hbar\right)\right), \tag{3.56}$$

where

$$\mathfrak{r}(z) = -\frac{1}{\hbar} \sum_{j \geq 1} \frac{(-1)^{j-1}}{j} \tau_j z^j. \tag{3.57}$$

One then has, by an explicit calculation,

$$\frac{\mathrm{d}}{\mathrm{d}y} \left( \mathcal{C}(z; y) \partial_\mathcal{C} R(y; \mathcal{C}(z; y)) \right) = -\frac{\partial R}{\partial y} - \mathcal{C} \frac{\partial_y R}{\partial_\mathcal{C} R} \frac{\partial^2 R}{\partial \mathcal{C}^2} + \mathcal{C} \frac{\partial^2 R}{\partial \mathcal{C} \partial y} = 0, \tag{3.58}$$

where we used that

$$\frac{\partial}{\partial y} \mathcal{C}(z; y) = -\frac{\partial_y R}{\partial_\mathcal{C} R}, \tag{3.59}$$

which follows from (3.52),

$$0 = \frac{\mathrm{d}}{\mathrm{d}y} R(y; \mathcal{C}(z; y)) = \frac{\partial R}{\partial y} + \frac{\partial \mathcal{C}}{\partial y} \frac{\partial R}{\partial \mathcal{C}}. \tag{3.60}$$

Let us now calculate $\mathsf{W}F$. We have

$$\mathsf{W}R(y; \mathcal{C}) = \frac{1}{\hbar^2} \left( \sum_{j \geq 1} (-1)^{j-1} \mathcal{C}^j \tau_j \mathrm{e}^{-j \mathrm{e}^{y\mathsf{D}} \mathcal{G}/\hbar} \right) \mathsf{W} \left( \mathrm{e}^{y\mathsf{D}} \mathcal{G} \right) = -\frac{1}{\hbar} \mathcal{C} \partial_\mathcal{C} R(y; \mathcal{C}) \, \mathsf{W} \left( \mathrm{e}^{y\mathsf{D}} \mathcal{G} \right). \tag{3.61}$$

We now note that all the dependence of $F$ on $\mathcal{G}$ and its derivatives is in the integration endpoint $R(y; \mathcal{C})$, since the integrand does not depend on $y$. Since $\mathsf{W}$ is a derivation, we simply have

$$\mathsf{W} \left( \int_0^{R(y; \mathcal{C})} \frac{z^m}{r(z)} \mathrm{d}z \right) = \frac{z^m}{r(z)} \bigg|_{z = R(y; \mathcal{C})} \mathsf{W}(R(y; \mathcal{C})) = -\frac{1}{\hbar} R^m(y; \mathcal{C}) \mathsf{W} \left( \mathrm{e}^{y\mathsf{D}} \mathcal{G} \right). \tag{3.62}$$

To proceed, we have to calculate the commutator of $\mathsf{W}$ with $\mathrm{e}^{y\mathsf{D}}$. This can be done with Hadamard's lemma,

$$\mathrm{e}^A B \mathrm{e}^{-A} = \sum_{n \geq 0} \frac{1}{n!} [A, B]_n, \tag{3.63}$$

where the iterated commutator $[A, B]_n$ is defined by

$$[A, B]_n = [A, [A, B]_{n-1}], \qquad [A, B]_0 = B. \tag{3.64}$$

In our case, and thanks to (3.32), we have the simple result that

$$[\mathsf{D}, \mathsf{W}]_{n \geq 1} = - (\mathsf{D}^n \mathcal{G}) \, \mathsf{D}, \qquad [\mathsf{D}, \mathsf{W}]_0 = \mathsf{W}, \tag{3.65}$$

therefore

$$\mathsf{W} \mathrm{e}^{y\mathsf{D}} = \mathrm{e}^{y\mathsf{D}} \left( \mathsf{W} - \sum_{k \geq 1} \frac{(-y)^k}{k!} \left( \mathsf{D}^k \mathcal{G} \right) \mathsf{D} \right) = \mathrm{e}^{y\mathsf{D}} \mathsf{W} + \left[ (\mathrm{e}^{y\mathsf{D}} - 1) \mathcal{G} \right] \mathrm{e}^{y\mathsf{D}} \mathsf{D}. \tag{3.66}$$

(A similar calculation appears in [29]). Taking now into account that $\mathsf{W}\mathcal{G} = 0$, we find

$$\mathsf{W}F = \sum_{m \geq 1} \frac{1}{m} [y^{m-1}] \left\{ R^m(y; \mathcal{C}) \left[ (\mathrm{e}^{y\mathsf{D}} - 1) \mathcal{G} \right] \left[ \mathrm{e}^{y\mathsf{D}} \mathsf{D} \mathcal{G} \right] \right\}. \tag{3.67}$$

We now note that

$$\left[(e^{yD} - 1)\mathcal{G}\right]\left[e^{yD}D\mathcal{G}\right] = \frac{1}{2}\frac{\partial H(y)}{\partial y}, \tag{3.68}$$

where

$$H(y) = \left[(e^{yD} - 1)\mathcal{G}\right]^2 \tag{3.69}$$

and $H(0) = 0$. Then, we can write

$$WF = \frac{1}{2}\sum_{m \geq 1}\frac{1}{m}[y^{m-1}]\left\{R^m(y;\mathcal{C})\frac{\partial H(y)}{\partial y}\right\}. \tag{3.70}$$

By using now the Lagrange inversion formula (3.48) we find,

$$WF = \frac{1}{2}\left[\sum_{k \geq 1}\frac{\Sigma^k(\mathcal{C})D^k\mathcal{G}}{k!}\right]^2. \tag{3.71}$$

Note that, since $\Sigma(\mathcal{C})$ does not commute with $D$, we have to expand in $y$ first, and replace $y$ by $\Sigma(\mathcal{C})$ afterwards.

Let us now evaluate $DF$:

$$DF = \sum_{m \geq 1}\frac{1}{m}[y^{m-1}]\left\{R^m(y;\mathcal{C})\left[e^{yD}D\mathcal{G}\right]\right\}. \tag{3.72}$$

We can write

$$e^{yD}D\mathcal{G} = \frac{\partial}{\partial y}h(y), \qquad h(y) = \left(e^{yD} - 1\right)\mathcal{G}, \tag{3.73}$$

with $h(0) = 0$, and by using again Lagrange inversion we obtain

$$DF = \sum_{k \geq 1}\frac{\Sigma^k(\mathcal{C})D^k\mathcal{G}}{k!}. \tag{3.74}$$

This shows that (3.51) satisfies (3.34), as we claimed.

Let us now calculate the initial condition satisfied by this solution. To do this, we first note that (3.51) leads to the formula

$$F^{(n)} = -\frac{\hbar}{n}\Sigma_n. \tag{3.75}$$

$\Sigma_n$ can be computed explicitly from (3.50) by using twice Faà di Bruno's formula. In this formula one has to sum over partitions, which are denoted by vectors $\boldsymbol{k} = (k_1, k_2, \cdots)$. We also denote

$$d(\boldsymbol{k}) = \sum_j jk_j, \qquad |\boldsymbol{k}| = \sum_j k_j. \tag{3.76}$$

One obtains,

$$\frac{1}{n!}\frac{d^n R^m(y;\mathcal{C})}{d\mathcal{C}^n}\bigg|_{\mathcal{C}=0} = \frac{m!(-1)^n}{\hbar^m}e^{-ne^{yD}\mathcal{G}/\hbar}\sum_{\substack{d(\boldsymbol{k})=n \\ |\boldsymbol{k}|=m}}\prod_j\frac{\tau_j^{k_j}}{j^{k_j}\,k_j!}, \tag{3.77}$$

as well as

$$\frac{1}{(m-1)!}\frac{\mathrm{d}^{m-1}}{\mathrm{d}y^{m-1}}\mathrm{e}^{-n\mathrm{e}^{y\mathsf{D}}\mathcal{G}/\hbar}\bigg|_{y=0} = \sum_{d(\boldsymbol{v})=m-1}\left(-\frac{n}{\hbar}\right)^{|\boldsymbol{v}|}\prod_j\frac{1}{v_j!}\left(\frac{\mathsf{D}^j\mathcal{G}}{j!}\right)^{v_j}. \tag{3.78}$$

Putting everything together, we find

$$F^{(n)} = \frac{\mathrm{e}^{-n\mathcal{G}/\hbar}}{n}\sum_{m=1}^{n}\frac{(-1)^{n-1}(m-1)!}{\hbar^{m-1}}\sum_{\substack{d(\boldsymbol{k})=n\\|\boldsymbol{k}|=m}}\sum_{d(\boldsymbol{v})=m-1}\left(-\frac{n}{\hbar}\right)^{|\boldsymbol{v}|}\prod_j\frac{\tau_j^{k_j}}{v_j!j^{k_j}k_j!}\left(\frac{\mathsf{D}^j\mathcal{G}}{j!}\right)^{v_j}. \tag{3.79}$$

From this explicit expression we can immediately obtain the value in the $\mathcal{A}$ frame: since $\mathsf{D}$ acts as zero on this frame, it is the term involving no derivatives $\mathsf{D}^j\mathcal{G}$. It corresponds to the empty partition $\boldsymbol{v}$, hence to $m=1$. The corresponding partition $\boldsymbol{k}$ is the $n$-dimensional vector $\boldsymbol{k}=(0,\cdots,0,1)$. One then finds,

$$F_{\mathcal{A}}^{(n)} = \frac{(-1)^{n-1}}{n^2}\tau_n\mathrm{e}^{-n\mathcal{A}/\hbar}. \tag{3.80}$$

We conclude that (3.79) gives the explicit, general formula for the multi-instanton amplitude $F^{(n)}$ that we promised at the beginning of this section.

**Example 3.1.** Let us list $F^{(n)}$, for the very first values of $n$. We have

$$\begin{aligned}
F^{(1)} &= \tau_1\mathrm{e}^{-\mathcal{G}/\hbar}, \\
F^{(2)} &= \left(-\frac{\tau_2}{4} + \tau_1^2\frac{\mathsf{D}\mathcal{G}}{2\hbar^2}\right)\mathrm{e}^{-2\mathcal{G}/\hbar}, \\
F^{(3)} &= \left\{\frac{\tau_3}{9} - \tau_1\tau_2\frac{\mathsf{D}\mathcal{G}}{2\hbar^2} + \tau_1^3\left(\frac{(\mathsf{D}\mathcal{G})^2}{2\hbar^4} - \frac{\mathsf{D}^2\mathcal{G}}{6\hbar^3}\right)\right\}\mathrm{e}^{-3\mathcal{G}/\hbar}.
\end{aligned} \tag{3.81}$$

It can be checked that (3.79) reproduces all the results obtained for low values of $n$ in [27, 28].

We now want to solve (3.40) with the boundary conditions (3.21). We claim that the solution is given by

$$w = w^{(0)} + \sum_{m\geq 1}\frac{1}{m}[y^{m-1}]R^m(y;\mathcal{C})\mathrm{e}^{y\mathsf{D}}\mathsf{D}w^{(0)}. \tag{3.82}$$

To check this statement, we first calculate

$$\begin{aligned}
\mathsf{W}w &= \sum_{m\geq 1}\frac{1}{m}[y^{m-1}]R^m(y;\mathcal{C})\mathsf{W}\left(\mathrm{e}^{y\mathsf{D}}\mathsf{D}w^{(0)}\right) \\
&+ \sum_{m\geq 1}[y^{m-1}]R^{m-1}(y;\mathcal{C})^{m-1}\mathsf{W}\left(R^m(y;\mathcal{C})\right)\mathrm{e}^{y\mathsf{D}}\mathsf{D}w^{(0)}
\end{aligned} \tag{3.83}$$

where we used $\mathsf{W}w^{(0)}=0$. By using (3.66), we get

$$\mathsf{W}\left(\mathrm{e}^{y\mathsf{D}}\mathsf{D}w^{(0)}\right) = \left(\mathrm{e}^{y\mathsf{D}}\mathsf{D}\mathcal{G}\right)\left(\mathrm{e}^{y\mathsf{D}}\mathsf{D}w^{(0)}\right) + \left[\left(\mathrm{e}^{y\mathsf{D}}-1\right)\mathcal{G}\right]\mathrm{e}^{y\mathsf{D}}\mathsf{D}^2w^{(0)}. \tag{3.84}$$

On the other hand

$$
\begin{aligned}
\mathsf{D}w = \mathsf{D}w^{(0)} &+ \sum_{m \geq 1} \frac{1}{m} [y^{m-1}] R^m(y;\mathcal{C}) \mathrm{e}^{y\mathsf{D}} \mathsf{D}^2 w^{(0)} \\
&+ \sum_{m \geq 1} [y^{m-1}] R^{m-1}(y;\mathcal{C}) \mathsf{D} \left( R(y;\mathcal{C}) \right) \mathrm{e}^{y\mathsf{D}} \mathsf{D}w^{(0)}.
\end{aligned}
\tag{3.85}
$$

Let us now write

$$
\begin{aligned}
\mathsf{D}F\mathsf{D}w = &\sum_{m \geq 1} \frac{1}{m} [y^{m-1}] \left\{ R^m(y;\mathcal{C}) \left[ \mathrm{e}^{y\mathsf{D}} \mathsf{D}\mathcal{G} \right] \mathsf{D}w^{(0)} \right\} \\
&+ \mathsf{D}F \sum_{m \geq 1} \frac{1}{m} [y^{m-1}] R^m(y;\mathcal{C}) \mathrm{e}^{y\mathsf{D}} \mathsf{D}^2 w^{(0)} \\
&+ \mathsf{D}F \sum_{m \geq 1} [y^{m-1}] R^{m-1}(y;\mathcal{C}) \mathsf{D} \left( R(y;\mathcal{C}) \right) \mathrm{e}^{y\mathsf{D}} \mathsf{D}w^{(0)}.
\end{aligned}
\tag{3.86}
$$

We can now calculate $\mathsf{W}w - \mathsf{D}F\mathsf{D}w$. The first line of (3.83) combines with minus the first line of (3.86) into

$$
\begin{aligned}
\sum_{m \geq 1} \frac{1}{m} [y^{m-1}] &\left\{ R^m(y;\mathcal{C}) \frac{\partial}{\partial y} \left( \left[ (\mathrm{e}^{y\mathsf{D}} - 1)\mathcal{G} \right] \left[ (\mathrm{e}^{y\mathsf{D}} - 1)\mathsf{D}w^{(0)} \right] \right) \right\} \\
&= \left[ \sum_{k=1}^{\infty} \frac{\Sigma^k(\mathcal{C})\mathsf{D}^k\mathcal{G}}{k!} \right] \left[ \sum_{k=0}^{\infty} \frac{\Sigma^k(\mathcal{C})\mathsf{D}^{k+1}w^{(0)}}{k!} \right].
\end{aligned}
\tag{3.87}
$$

The second line of (3.86), once subtracted, gives

$$
-\mathsf{D}F \sum_{m \geq 1} \frac{1}{m} [y^{m-1}] R^m(y;\mathcal{C}) \frac{\partial}{\partial y} \left( \left[ \mathrm{e}^{y\mathsf{D}} - 1 \right] \mathsf{D}w^{(0)} \right) = -\mathsf{D}F \left[ \sum_{k=0}^{\infty} \frac{\Sigma^k(\mathcal{C})\mathsf{D}^{k+1}w^{(0)}}{k!} \right].
\tag{3.88}
$$

This cancels (3.87), after using (3.74). It remains to prove that the last line in (3.83) cancels against the last line in (3.86). To show this, we notice that by direct calculation one finds

$$
\mathsf{W} \left( R(y;\mathcal{C}) \right) = \left[ \left( \mathrm{e}^{y\mathsf{D}} - 1 \right) \mathcal{G} \right] \mathsf{D} \left( R(y;\mathcal{C}) \right).
\tag{3.89}
$$

Therefore, we have to show that

$$
\begin{aligned}
\sum_{m \geq 0} [y^m] R^m(y;\mathcal{C}) \mathsf{D} \left( R(y;\mathcal{C}) \right) &\left[ \left( \mathrm{e}^{y\mathsf{D}} - 1 \right) \mathcal{G} \right] \mathrm{e}^{y\mathsf{D}} \mathsf{D}w^{(0)} \\
&= \mathsf{D}F \sum_{m \geq 1} [y^{m-1}] R^{m-1}(y;\mathcal{C}) \mathsf{D} \left( R(y;\mathcal{C}) \right) \mathrm{e}^{y\mathsf{D}} \mathsf{D}w^{(0)}.
\end{aligned}
\tag{3.90}
$$

This follows from (3.49) and from the explicit value of $\mathsf{D}F$ in (3.74), since both sides evaluate to the same power series, namely

$$
\frac{1}{1 - \partial_y R(\Sigma(\mathcal{C});\mathcal{C})} \mathsf{D} \left( R(\Sigma(\mathcal{C});\mathcal{C}) \right) \left[ \sum_{k=1}^{\infty} \frac{\Sigma^k(\mathcal{C})\mathsf{D}^k\mathcal{G}}{k!} \right] \left[ \sum_{k=0}^{\infty} \frac{\Sigma^k(\mathcal{C})\mathsf{D}^{k+1}w^{(0)}}{k!} \right].
\tag{3.91}
$$

The proof is now complete.

From (3.113) it is possible to obtain an explicit solution for the instanton corrections to the Wilson loop, as follows

$$w^{(n)} = \sum_{d(\boldsymbol{k})=n} \mathsf{D}^{|\boldsymbol{k}|} w^{(0)} \prod_j \frac{(\Sigma_j)^{k_j}}{k_j!}. \tag{3.92}$$

From this formula it is manifest that $w_{\mathcal{A}}^{(n)} = 0$, since its expression always involves actions of $\mathsf{D}$. If we write

$$w^{(n)} = \Omega_n \mathrm{e}^{-n\mathcal{G}/\hbar}, \tag{3.93}$$

we obtain for example

$$\Omega_1 = -\frac{\tau_1}{\hbar} \mathsf{D} w^{(0)},$$

$$\Omega_2 = \frac{\tau_2}{2\hbar} \mathsf{D} w^{(0)} + \tau_1^2 \left( \frac{\mathsf{D}^2 w^{(0)}}{2\hbar^2} - \frac{\mathsf{D}\mathcal{G}\mathsf{D} w^{(0)}}{\hbar^3} \right),$$

$$\Omega_3 = -\frac{\tau_3}{3\hbar} \mathsf{D} w^{(0)} + \tau_1^3 \left( \frac{\mathsf{D}^2 \mathcal{G}\mathsf{D} w^{(0)}}{2\hbar^4} - \frac{3(\mathsf{D}\mathcal{G})^2 \mathsf{D} w^{(0)}}{2\hbar^5} + \frac{\mathsf{D}\mathcal{G}\mathsf{D}^2 w^{(0)}}{\hbar^4} - \frac{\mathsf{D}^3 w^{(0)}}{6\hbar^3} \right) \tag{3.94}$$

$$- \frac{\tau_1 \tau_2}{2} \left( \frac{3\mathsf{D}\mathcal{G}\mathsf{D} w^{(0)}}{\hbar^3} - \frac{\mathsf{D}^2 w^{(0)}}{\hbar^2} \right).$$

So far we have obtained explicit multi-instanton amplitudes $F^{(\ell)}$, $w^{(\ell)}$ based on the ansätze (3.1), (3.17). However, the HAE admit a more general solution in terms of a two-parameter trans-series. This is because both $F^{(0)}$ and $w^{(0)}$ are formal power series in $\hbar^2$, and therefore the singularities of their Borel transform (which correspond to instanton actions) come in pairs. At the level of trans-series, this means that, if we have a trans-series solution with an exponential of the form $\mathrm{e}^{-\mathcal{A}/\hbar}$, there should be an "anti-instanton" amplitude involving the opposite exponential $\mathrm{e}^{\mathcal{A}/\hbar}$. The paradigmatic example of this phenomenon occurs in the Painlevé I equation describing 2d gravity. The general trans-series solution to this equation was studied in [50] and it involves both instantons and "anti-instantons," as well as mixed sectors (see [51–54] for further studies of this type of trans-series solutions to non-linear ODEs, and [55] for a recent matrix model interpretation). We will then consider a more general ansatz, of the form

$$F = \sum_{n,m \geq 0, (n,m) \neq (0,0)} \mathcal{C}_1^n \mathcal{C}_2^m F^{(n|m)}, \tag{3.95}$$

where we assume that

$$F^{(n|m)} \sim \exp\left( -\frac{(n-m)\mathcal{A}}{\hbar} \right). \tag{3.96}$$

Since the perturbative series are even in $\hbar$, the solutions of the HAE should respect the following symmetry under $\hbar \to -\hbar$

$$F^{(0|m)}(\hbar) = F^{(m|0)}(-\hbar). \tag{3.97}$$

We will then consider the boundary conditions

$$F_{\mathcal{A}}^{(\ell|0)} = \tau_\ell \frac{(-1)^{\ell-1}}{\ell^2} \mathrm{e}^{-\ell\mathcal{A}/\hbar}, \qquad F_{\mathcal{A}}^{(0|\ell)} = \tau_\ell \frac{(-1)^{\ell-1}}{\ell^2} \mathrm{e}^{\ell\mathcal{A}/\hbar}, \tag{3.98}$$

and all the other sectors vanish in the $\mathcal{A}$ frame. Similar considerations apply to the Wilson loops, and we have to consider more general "mixed" instanton solutions $w^{(n|m)}$. In this case, the boundary conditions are simply $w^{(n|m)}_{\mathcal{A}} = 0$.

We have obtained an explicit formula for $F^{(n|0)}$ in (3.79), but we do not have closed form expressions for the general mixed sectors $F^{(n|m)}$. However, they can be calculated systematically by using the algebra of operators, as it was done in [27, 28] to calculate $F^{(n|0)}$. One writes the $F^{(n|m)}$ as linear combinations of "words" of the form

$$(\mathsf{D}\mathcal{G})^{k_1}(\mathsf{D}^2\mathcal{G})^{k_2}\cdots \tag{3.99}$$

labelled by a vector $\boldsymbol{k}$ satisfying $d(\boldsymbol{k}) \leq n + m - 1$. In the case of $w^{(n|m)}$, we consider words of the form

$$\mathsf{D}^s w^{(0)}(\mathsf{D}\mathcal{G})^{k_1}(\mathsf{D}^2\mathcal{G})^{k_2}\cdots , \tag{3.100}$$

where $1 \leq s \leq n + m$ and $d(\boldsymbol{k}) \leq n + m - s$. By using the commutation relation (3.32) and $\mathsf{W}\mathcal{G} = \mathsf{W}w^{(0)} = 0$, one can solve the equations for the mixed sectors systematically.

As an example, let us consider the first mixed sector $(1|1)$. The instanton free energy satisfies the equation

$$\mathsf{W}F^{(1|1)} = \mathsf{D}F^{(1|0)}\mathsf{D}F^{(0|1)}, \tag{3.101}$$

which can be solved to give

$$F^{(1|1)} = -\frac{\tau_1^2}{\hbar^2}\mathsf{D}\mathcal{G}. \tag{3.102}$$

We also find, for the corresponding Wilson loop,

$$w^{(1|1)} = -\frac{\tau_1^2}{\hbar^2}\mathsf{D}^2 w^{(0)}. \tag{3.103}$$

We will now consider a special family of solutions to the HAE which play an important rôle in the construction. They are labelled by a positive integer $\ell \in \mathbb{Z}_{>0}$, and they are defined by the boundary conditions (3.98) with the particular choice

$$\tau_k = \delta_{k\ell}. \tag{3.104}$$

We will denote the corresponding solutions by $F^{(n|m)}_\ell$, $w^{(n|m)}_\ell$. It is easy to check, from the explicit expressions (3.79) and (3.92), that

$$F^{(\ell)}_\ell = \frac{(-1)^{\ell-1}}{\ell^2}\mathrm{e}^{-\ell\mathcal{G}/\hbar}, \tag{3.105}$$

for the free energies, and

$$w^{(\ell)}_\ell = \frac{1}{\hbar}\frac{(-1)^\ell}{\ell}\mathsf{D}w^{(0)}\mathrm{e}^{-\ell\mathcal{G}/\hbar} \tag{3.106}$$

for the Wilson loops. We also have that $F^{(n)}_\ell = w^{(n)}_\ell = 0$ unless $n = k\ell$, $k = 1, 2, \cdots$. As we will see, these solutions are the relevant ones in the actual calculation of alien derivatives. We also note that a very similar structure appears in the conventional topological string [29].

The trans-series solution for the holomorphic limit of the Wilson loop leads to a trans-series solution for the quantum $A$-period, by simply extending the implicit equation (2.31) to a trans-series version. We consider the ansatz,

$$t = \sum_{n \geq 0} \mathcal{C}^n t^{(n)}, \tag{3.107}$$

where $t^{(0)} = t(z;\hbar)$ is the quantum $A$-period. We then have the following trans-series equation,

$$\omega\left(\sum_{n\geq 0}\mathcal{C}^n t^{(n)}\right) = \omega^{(0)}(z(t^{(0)})) = \log z, \tag{3.108}$$

where $\omega$ is the holomorphic limit of the trans-series (3.17). Note that, as indicated in (2.31), the functions $t^{(n)}$ on the left hand side are evaluated not at $z$, but at $z(t(z;\hbar))$. Solving this order by order in $\mathcal{C}$, one can find explicit expressions for the $t^{(n)}$. We find for example,

$$t^{(1)} = -\frac{\omega^{(1)}}{\partial_t \omega^{(0)}} = \frac{\alpha}{\hbar}\tau_1 e^{-\mathcal{G}/\hbar}, \tag{3.109}$$

where we have taken into account (3.27). One can do better and use the results above to solve (3.108) in closed form. To do this, we consider the holomorphic limit of the solutions found above. Due to (3.27), the holomorphic version of $R(y;\mathcal{C})$ involves a function of a shifted argument. Let us introduce

$$\mathcal{R}(t;\mathcal{C}) = -\frac{1}{\hbar}\sum_{j\geq 1}\mathcal{C}^j \frac{(-1)^{j-1}}{j}\tau_j \exp\left(-\frac{j}{\hbar}\mathcal{G}(t)\right). \tag{3.110}$$

Then, the holomorphic limit of $R(y;\mathcal{C})$ is

$$R(y;\mathcal{C}) \to \mathcal{R}(t+\alpha y;\mathcal{C}) \tag{3.111}$$

and the holomorphic limit of $\Sigma(\mathcal{C})$ is a function of $t$ which solves the implicit equation

$$\Sigma(t;\mathcal{C}) = \mathcal{R}\left(t+\alpha\Sigma(t;\mathcal{C});\mathcal{C}\right). \tag{3.112}$$

Finally, by using Lagrange inversion (3.48), the solution (3.82) for the Wilson loop becomes in the holomorphic limit

$$w(t) = w^{(0)}\left(t+\alpha\Sigma(t;\mathcal{C})\right). \tag{3.113}$$

If we now introduce

$$t^{(0)} = t+\alpha\Sigma(t;\mathcal{C}) \tag{3.114}$$

we can write (3.113) as

$$w\left(t^{(0)} - \alpha\mathcal{R}(t^{(0)};\mathcal{C})\right) = w^{(0)}(t^{(0)}). \tag{3.115}$$

By comparing this result to (3.108), we conclude that the quantum $A$-period trans-series is simply

$$\sum_{n\geq 1}\mathcal{C}^n t^{(n)} = -\alpha R(t^{(0)};\mathcal{C}), \tag{3.116}$$

in other words,

$$t^{(n)} = \frac{\alpha}{\hbar}\frac{(-1)^{n-1}}{n}\tau_n e^{-n\mathcal{G}(t^{(0)})/\hbar}. \tag{3.117}$$

In particular, the trans-series corresponding to the discrete family of solutions considered in (3.105), (3.106) is

$$t_\ell^{(\ell)} = \frac{\alpha}{\hbar}\frac{(-1)^{\ell-1}}{\ell}e^{-\ell\mathcal{G}(t^{(0)})/\hbar}. \tag{3.118}$$

We note that the r.h.s. of (3.117), (3.118) involves the holomorphic limit of $\mathcal{G}$ evaluated at $t^{(0)}$, i.e. $\mathcal{G}(z(t(z;\hbar));\hbar)$, and it is therefore a composition of two formal series in $\hbar$. This is in contrast to (3.105), (3.106), which involve simply $\mathcal{G}(z;\hbar)$. We also note that, with the boundary conditions (3.98), one has $t^{(n|m)} = 0$, unless $m = 0$ or $n = 0$.

## 3.3 From multi-instantons to the resurgent structure

The resurgent structure associated to the quantum periods consists of their Borel singularities, the trans-series associated to these singularities, and the corresponding Stokes coefficients. A very useful way to encode this information is through the language of alien derivatives [5]. Alien derivatives are labelled by singularities in the Borel plane, and give essentially the trans-series associated to that singularity, together with the Stokes coefficient. We refer to the companion paper [29] for a lightning review of alien calculus and some references on the subject. We also mention that the resurgent structure of the NS free energy of the resolved conifold has been recently studied in [56–58].

Let $\mathcal{F}_n^{(0)}(z)$, $\omega_n^{(0)}(z)$ be the holomorphic limits of the NS free energies and Wilson loops, in a given frame. The Borel transforms

$$\widehat{\mathcal{F}}^{(0)}(z,\zeta) = \sum_{n \geq 0} \frac{1}{(2n)!} \mathcal{F}_n^{(0)}(z)\zeta^{2n}, \qquad \widehat{\omega}^{(0)}(z,\zeta) = \sum_{n \geq 0} \frac{1}{(2n)!}\omega_n^{(0)}(z)\zeta^{2n} \tag{3.119}$$

will have singularities filling subsets of a lattice in the complex plane. Let us consider a singularity $\ell\mathcal{A}$, where $\mathcal{A}$ is a "primitive" singularity of the form (3.3), and $\ell \in \mathbb{Z}_{>0}$ is a positive integer. There are two different cases to consider. If $\alpha = 0$, i.e. if the instanton action is given by the flat coordinate of the frame (up to a linear shift by a constant), then the multi-instanton trans-series are trivial, and of the form (3.14). In this case, we will have

$$\dot{\Delta}_{\ell\mathcal{A}}\mathcal{F}^{(0)} = \frac{1}{\hbar}\mathsf{S}_{\ell\mathcal{A}}^F(\hbar)\mathcal{F}_{\mathcal{A}}^{(\ell)}. \tag{3.120}$$

Here, $\mathsf{S}_{\mathcal{A}}(\hbar)$ is a Stokes coefficient which depends on $\hbar$ and in principle also on the modulus $t$. Our concrete calculations indicate that the dependence on $\hbar$ is simple and that they are locally constant functions of $t$. We also expect the Stokes constants to be independent of $\ell$ in many cases, as suggested by the large order behavior (3.11). We conjecture in addition that, when $\alpha = 0$,

$$\dot{\Delta}_{\ell\mathcal{A}}\omega^{(0)} = 0. \tag{3.121}$$

Let us now consider the more interesting case $\alpha \neq 0$, in which instanton sectors are non-trivial. We conjecture the following result for the pointed alien derivatives,

$$\dot{\Delta}_{\ell\mathcal{A}}\mathcal{F}^{(0)} = \frac{1}{\hbar}\mathsf{S}_{\ell\mathcal{A}}^F(\hbar)\mathcal{F}_\ell^{(\ell)},$$
$$\dot{\Delta}_{\ell\mathcal{A}}\omega^{(0)} = \mathsf{S}_{\ell\mathcal{A}}^\omega(\hbar)\omega_\ell^{(\ell)}, \tag{3.122}$$

where $\mathcal{F}_\ell^{(\ell)}$, $\omega_\ell^{(\ell)}$ are the holomorphic limits of (3.105), (3.106), respectively. $\mathsf{S}_{\ell\mathcal{A}}^{F,\omega}(\hbar)$ are Stokes coefficients with the properties noted above. In addition, our calculations suggest that

$$\mathsf{S}_{\ell\mathcal{A}}^F(\hbar) = \mathsf{S}_{\ell\mathcal{A}}^\omega(\hbar) =: \mathsf{S}_{\ell\mathcal{A}}(\hbar), \tag{3.123}$$

which we will assume always to be the case. Since $\widetilde{\mathcal{F}}^{(0)}$, $\omega^{(0)}$ are series in even powers of $\hbar$, we have

$$\dot{\Delta}_{-\ell\mathcal{A}}\mathcal{F}^{(0)} = \frac{1}{\hbar}\mathsf{S}_{\ell\mathcal{A}}(-\hbar)\mathcal{F}_\ell^{(0|\ell)},$$
$$\dot{\Delta}_{-\ell\mathcal{A}}\omega^{(0)} = \mathsf{S}_{\ell\mathcal{A}}(-\hbar)\omega_\ell^{(0|\ell)}. \tag{3.124}$$

The conjecture (3.122) is motivated by the large order behavior (3.11) and we will give empirical evidence for it in the next section. In fact, from these examples we find the Stokes constants $\mathsf{S}_{\ell\mathcal{A}}(\hbar)$ are independent of $\ell$ in many situations. We would like to also emphasize that (3.120) and (3.121) can be regarded as special cases of (3.122) evaluated in a special frame where $\mathcal{A}$ up to a constant is the A-period, and that the Stokes constants do not depend on the frame. We note that a similar conjecture applies to the topological string free energy [29]. The Stokes coefficients appearing in the above equations encode information about the resurgent structure of the theory and they are non-trivial. Currently, we can only calculate them numerically, and we have access to very few of them.

The alien derivatives (3.122) are the "primitive" ones, in the sense that based on them all additional alien derivatives can be calculated. This is simply because all multi-instanton amplitudes are functionals of $\mathcal{F}^{(0)}$ and $\omega^{(0)}$, and the action of alien derivatives commutes with taking derivatives w.r.t. $t$. In particular, since $\mathcal{G} = \hbar\mathsf{D}\mathcal{F}^{(0)}$, we have

$$\dot{\Delta}_{\ell\mathcal{A}}\mathcal{G} = \frac{1}{\hbar}\mathsf{S}_{\ell\mathcal{A}}(\hbar)\frac{(-1)^{\ell}}{\ell}\mathsf{D}\mathcal{G}\,\mathrm{e}^{-\ell\mathcal{G}/\hbar}. \tag{3.125}$$

Assuming that (3.123) holds, we have the following formulae for the alien derivatives of the full family of trans-series:

$$\dot{\Delta}_{\ell\mathcal{A}}\mathcal{F}_{\ell}^{(\ell n|\ell m)} = \frac{1}{\hbar}\mathsf{S}_{\ell\mathcal{A}}(\hbar)(n+1)\mathcal{F}_{\ell}^{(\ell(n+1)|\ell m)},$$

$$\dot{\Delta}_{\ell\mathcal{A}}\omega_{\ell}^{(\ell n|\ell m)} = \mathsf{S}_{\ell\mathcal{A}}(\hbar)(n+1)\omega_{\ell}^{(\ell(n+1)|\ell m)}. \tag{3.126}$$

One also finds,

$$\dot{\Delta}_{-\ell\mathcal{A}}\mathcal{F}_{\ell}^{(\ell n|\ell m)} = \frac{1}{\hbar}\mathsf{S}_{\ell\mathcal{A}}(-\hbar)(m+1)\mathcal{F}_{\ell}^{(\ell n|\ell(m+1))},$$

$$\dot{\Delta}_{-\ell\mathcal{A}}\omega_{\ell}^{(\ell n|\ell m)} = \mathsf{S}_{\ell\mathcal{A}}(-\hbar)(m+1)\omega_{\ell}^{(\ell n|\ell(m+1))}. \tag{3.127}$$

These expressions follow directly from (3.122) by taking derivatives, and although we don't have a general proof, we have checked them in many cases. They are very similar to the equations for alien derivatives which follow from bridge equations in Écalle's theory of ODEs (see e.g. [59, 60]). In fact, by using that the alien derivatives commute with the operators $\mathsf{W}$, $\mathsf{D}$, one can further justify (3.126), see [29] for a similar argument in the case of the conventional topological string.

We can now ask what is the value of the alien derivatives for the quantum A-period. Like before, the answer depends on the value of $\alpha$ in (3.3). If $\alpha = 0$, we have

$$\dot{\Delta}_{\ell\mathcal{A}}t^{(0)} = 0. \tag{3.128}$$

If $\alpha \neq 0$, we expect them to be given by the same family of solutions (3.118) that appears in the alien derivatives (3.122), namely

$$\dot{\Delta}_{\ell\mathcal{A}}t^{(0)} = \mathsf{S}_{\ell\mathcal{A}}(\hbar)t_{\ell}^{(\ell)}. \tag{3.129}$$

(3.128) and (3.129) can be derived from the formulae above for the alien derivatives by using the following general result [6, 7][3]. Let us consider a formal power series

$$\varphi(x, \hbar) = \sum_{n \geq 0} a_n(x)\hbar^n, \tag{3.130}$$

---

[3]We would like to thank David Sauzin for important clarifications concerning this result.

where $a_n(x)$ are analytic functions of $x$ at $x = 0$. Let us now suppose that we replace $x$ by a formal power series $\psi(\hbar)$, to obtain the formal power series

$$\rho(\hbar) = \varphi\left(\psi(\hbar), \hbar\right). \tag{3.131}$$

Then, we have the following formula for the pointed alien derivative,

$$\dot{\Delta}_{\mathcal{A}}\rho(\hbar) = (\dot{\Delta}_{\mathcal{A}}\varphi)(\psi(\hbar), \hbar) + \left.\frac{\partial}{\partial x}\varphi(x, \hbar)\right|_{x=\psi(\hbar)} \dot{\Delta}_{\mathcal{A}}\psi(\hbar), \tag{3.132}$$

where $\mathcal{A}$ is arbitrary. This formula makes it possible to calculate the alien derivatives of the quantum $A$-period from those of the Wilson loop. By applying the alien derivative to (2.31), we find

$$\dot{\Delta}_{\ell\mathcal{A}}\omega(z(t(z;\hbar)); \hbar) = \dot{\Delta}_{\ell\mathcal{A}}\log(z) = 0, \tag{3.133}$$

and by using (3.132) in the first term, we obtain

$$\dot{\Delta}_{\ell\mathcal{A}}t(z;\hbar) = -\frac{1}{\frac{\partial}{\partial t}\omega(z(t(z;\hbar)); \hbar)}(\dot{\Delta}_{\ell\mathcal{A}}\omega)(z(t(z;\hbar)); \hbar). \tag{3.134}$$

Note that, in particular, the Borel singularities of $t(z;\hbar)$ in the Borel plane of $\hbar$, as a function of $z$, are the same ones as for $\omega(z;\hbar)$. It is now clear that (3.128) follows from (3.121) and (3.134). Taking into account (3.27), we conclude that (3.129) follows from (3.122) and (3.134).

We are now ready to derive the Delabaere–Pham formula from the above considerations. Like before, we will assume (3.123), as well as that the Stokes constants $\mathsf{S}_{\ell\mathcal{A}}$ are in fact independent of $\ell$ as seen in many situations and also checked with examples in section 4,

$$\mathsf{S}_{\ell\mathcal{A}}(\hbar) = \mathsf{S}_{\mathcal{A}}(\hbar), \quad \forall \ell. \tag{3.135}$$

Let us suppose that $\ell\mathcal{A}$ is a Borel singularity common to $\mathcal{F}^{\mathrm{NS}}$ and to the quantum period $t$. According to our conjecture, this happens if $\alpha \neq 0$ in (3.3). We first calculate the alien derivative of $\mathcal{G}(t^{(0)}) = \mathcal{G}(z(t(z;\hbar))$ at $\ell\mathcal{A}$. We have,

$$\dot{\Delta}_{\ell\mathcal{A}}\mathcal{G}(t^{(0)}) = \left(\dot{\Delta}_{\ell\mathcal{A}}\mathcal{G}\right)(t^{(0)}) + \frac{\partial \mathcal{G}}{\partial t}\dot{\Delta}_{\ell\mathcal{A}}t(z;\hbar) = 0, \tag{3.136}$$

where we used first (3.132), and then we used the values of the alien derivatives (3.125) and (3.129), respectively. Let us now compute the Stokes automorphism $\mathfrak{S}$ in the direction where lie the singularities $\ell\mathcal{A}$, $\ell = 1, 2, \cdots$. We recall that the Stokes automorphism is the exponent of the (dotted) alien derivatives [5, 6, 60, 61]

$$\mathfrak{S} = \exp\left(\sum_{\ell=1}^{\infty} \dot{\Delta}_{\ell\mathcal{A}}\right). \tag{3.137}$$

From (3.129) we read,

$$\dot{\Delta}_{\ell\mathcal{A}}t^{(0)} = \frac{\alpha}{\hbar}\mathsf{S}_{\mathcal{A}}(\hbar)\frac{(-1)^{\ell-1}}{\ell}\mathrm{e}^{-\ell\mathcal{G}(t^{(0)})/\hbar}, \tag{3.138}$$

and when more than one alien derivative acts on $t^{(0)}$ we get zero, due to (3.136). We conclude that

$$(\mathfrak{S} - 1)\, t^{(0)} = \frac{\alpha}{\hbar}\mathsf{S}_{\mathcal{A}}(\hbar)\log\left(1 + \mathrm{e}^{-\mathcal{G}(t^{(0)})/\hbar}\right), \tag{3.139}$$

which is the Delabaere–Pham formula. Interestingly, in this derivation our basic assumption is the first equation in (3.120), which selects the family of solutions (3.105) as the relevant one to compute alien derivatives. In turn, this family is suggested by the behavior at the conifold (3.11), as we have remarked many times. Therefore, this behavior, together with the HAE, goes a long way towards explaining the simplicity of the Delabaere–Pham formula. It is important to note however that such a simple formula does *not* apply to the free energies or the Wilson loop vevs, since multiple actions of the alien derivatives do not give zero. This is because in the equation (3.129), the r.h.s. involves a composition of formal series $\mathcal{G}(z(t(z;\hbar)))$, while the r.h.s. of (3.122) involves $\mathcal{G}(z)$, as we noted before.

Although our discussion has focused on the quantum $A$-period, similar considerations apply to the quantum $B$-period, by simply exchanging the rôle of $t$ and $\partial_t F_0$.

## 4 Examples and experimental evidence

### 4.1 The double well potential

The symmetric double well describes a particle in the potential

$$V(x) = \frac{x^2}{2}\left(1 + gx\right)^2.\tag{4.1}$$

This can be made symmetric w.r.t. $x = 0$ by shifting $x \to x - 1/(2g)$. The parameter $z$ is in this case the energy $\xi$ appearing in the Schrödinger equation.

In [31] it was shown that the quantum $A$- and $B$-periods, or the all-orders WKB periods of this quantum mechanical model, were related to each other by the quantum free energy, which is the analog of the NS free energy in supersymmetric gauge theories and topological string theories, and the quantum free energy is governed by the holomorphic anomaly equations (2.23). Similarly, one can also define the NS Wilson loop, which is in turn governed by the holomorphic anomaly equations (2.34). We collect all the necessary ingredients of quantum periods, free energies, and Wilson loops in Appendix A.

The double well quantum mechanical model has no large radius frame, but has two conifold frames, called the $\xi$-frame and the $\xi_D$-frame, associated respectively to the conifold point $\xi = 0$ and the dual conifold point $\xi = 1/32$. We will thus focus on the region of moduli space

$$0 \leq \xi \leq \frac{1}{32},\tag{4.2}$$

which in fact corresponds to a positive energy below the barrier. We first focus on the $\xi$-frame. The flat coordinate is

$$t = \xi\,_2F_1\left(\frac{1}{4}, \frac{3}{4}, 2, 32\xi\right),\tag{4.3}$$

and the dual classical period is

$$t_D = 2\sqrt{2}\pi\left(\frac{1}{32} - \xi\right)\,_2F_1\left(\frac{1}{4}, \frac{3}{4}; 2; 1 - 32\xi\right),\tag{4.4}$$

which is related to the prepotential by

$$t_D = \frac{\partial F_0(t)}{\partial t}.\tag{4.5}$$

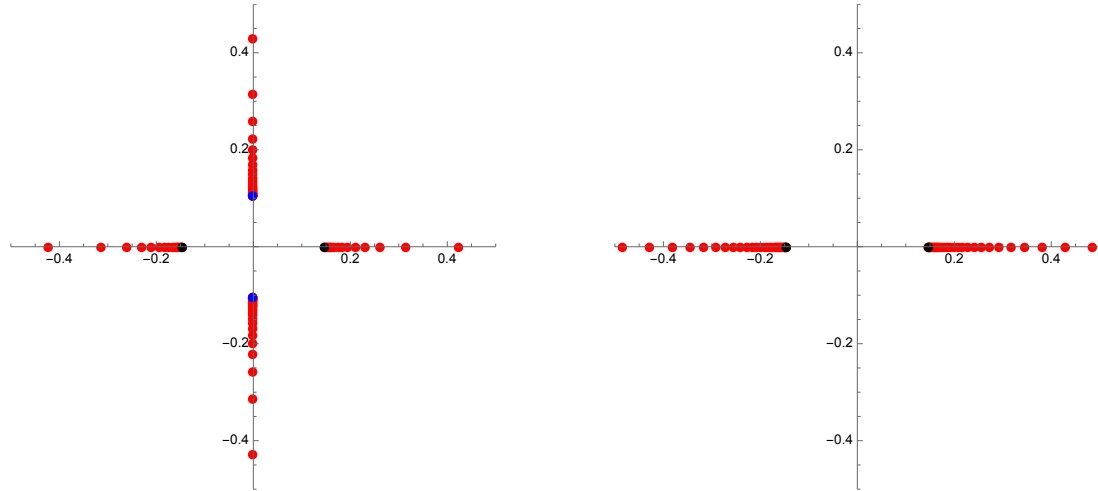

**Figure 1**: The Borel singularities for $\mathcal{F}^{\text{NS}}(\xi; \hbar)$ (left) and $\omega(\xi; \hbar)$ (right) at $\xi = 1/64$ in the $\xi$-frame. In both cases perturbative expansions up to $n = 145$ are used. The blue dots in the imaginary axis show the location of $\pm\mathcal{A}_t$. The black dots in the real axis show the location of $\pm\mathcal{A}_c$.

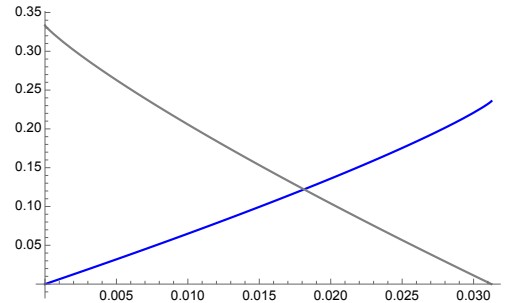

**Figure 2**: The value of $\text{Im}\mathcal{A}_t$ (blue) and $\text{Re}\mathcal{A}_c$ (gray) in the range $0 \le \xi \le 1/32$.

We find that the Borel singularities of the NS free energies are located at

$$m\mathcal{A}_t, \quad \ell\mathcal{A}_c, \qquad m, \ell \in \mathbb{Z}_{\neq 0}, \tag{4.6}$$

with

$$\mathcal{A}_t = 2\pi\mathrm{i}t(\xi), \quad \mathcal{A}_c = t_D(\xi) \tag{4.7}$$

while the Borel singularities of the NS Wilon loops are located at

$$\ell\mathcal{A}_c \qquad \ell \in \mathbb{Z}_{\neq 0}. \tag{4.8}$$

A good graphical guide to the location of Borel singularities is to consider truncations of the Borel transforms (3.119), obtained by keeping a large number of terms. Then, one looks at the poles of the diagonal Padé approximants to these polynomials. These poles mimick the location of branch cuts. As an example, we show in Fig. 1 the singularities of the Padé approximants of $\mathcal{F}^{\text{NS}}$ and $\omega$ in the Borel plane at $\xi = 1/64$. In the left panel, the branch points $\pm\mathcal{A}_t, \pm\mathcal{A}_c$ of $\mathcal{F}^{\text{NS}}$ are clearly visible, while in the right panel, one only sees the branch points $\pm\mathcal{A}_c$ for $\omega$.

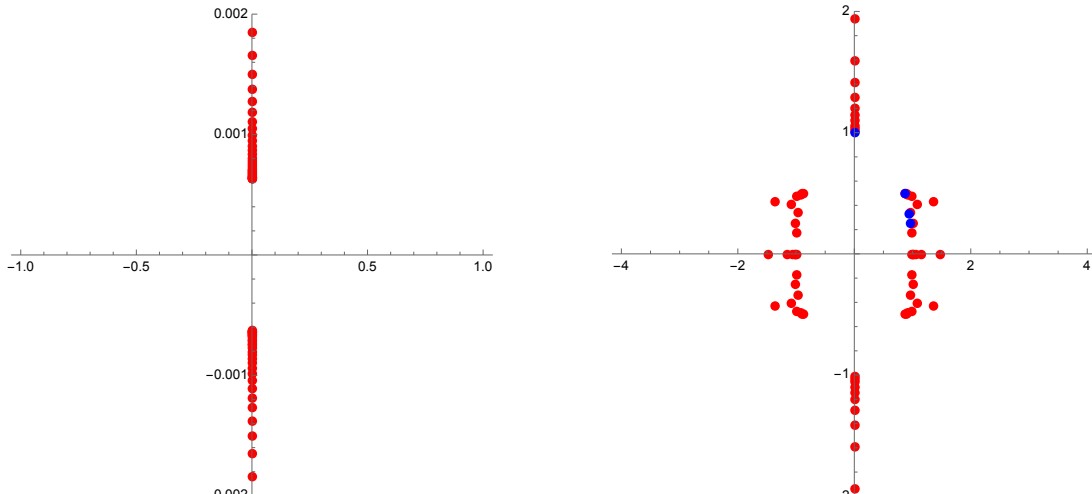

**Figure 3**: The Borel singularities for $\mathcal{F}^{\mathrm{NS}}(\xi;\hbar)$ at $\xi = 10^{-4}$ in the $\xi$-frame. In the left panel we plot the poles of the conventional Padé approximant in the $\zeta$-plane, while in the right panel we plot the poles of the conformal Padé approximant in the $\tau$-plane. In both cases we use the NS free energies up to $n = 145$. The blue dots correspond to the location of the singularities $m\mathcal{A}_c$ with $m = 1, 2, 3, 4$ in the $\tau$-plane.

Only leading order branch points are visible in Fig. 1, and higher order singularities are hidden behind the branch cuts. To unveil more singularities in the set (4.6), we first notice that in the range (4.2), $\mathcal{A}_t \in i\mathbb{R}_{\geq 0}$ and $\mathcal{A}_c \in \mathbb{R}_{\geq 0}$, $\mathcal{A}_t$ and $\mathcal{A}_c$ then compete with each other in strenghth in the range (4.2). As shown in Fig. 2, close to $\xi = 0$, $\pm\mathcal{A}_t$ is smaller in strength and thus are the more dominant Borel singularities of $\mathcal{F}^{\mathrm{NS}}$. For instance at $\xi = 10^{-4}$, one only finds the branch points $\pm\mathcal{A}_t$ in the Borel plane in the left panel of Fig. 3 (the less dominant singularities $\pm\mathcal{A}_c$ are difficult to detect as only a limited number of terms is used in the calculation of Borel transform). However by combining the Borel transform (3.119) with the conformal map

$$\zeta = \frac{1}{i}\frac{2\mathcal{A}_t\tau}{1 - \tau^2}, \tag{4.9}$$

higher order singularities $\pm m\mathcal{A}_t$ with $m = 2, 3, 4, \ldots$ are also visible in the Padé approximant in the $\tau$-plane, as shown in the right panel of Fig. 3.

Simiarly, close to $\xi = 1/32$, $\pm\mathcal{A}_c$ are the more dominant Borel singularities of $\mathcal{F}^{\mathrm{NS}}$. If we take $\xi = 1/32 - 10^{-4}$, we only find the branch points $\pm\mathcal{A}_c$ in the Borel plane in the left panel of Fig. 4. Nevertheless, by combining the Borel transform (3.119) with the conformal map

$$\zeta = \frac{2\mathcal{A}_c\tau}{1 + \tau^2}, \tag{4.10}$$

higher order singularities $\pm\ell\mathcal{A}_c$ with $\ell = 2, 3, 4, \ldots$ are also visible in the Padé approximant in the $\tau$-plane, as shown in the right panel of Fig. 4.

The Stokes coefficients associated to these singularities can be numerically calculated with high precision. To calculate the Stokes coefficients at $m\mathcal{A}_c$ of the NS free energies, we can choose $\xi$ very close to $\xi = 0$, for instance at $\xi = 10^{-4}$, so that $\pm\mathcal{A}_t$ are the dominant Borel singularities, as in the left panel of Fig. 3. We find then

$$(s_+ - s_-)\mathcal{F}^{(0)} = \frac{i\hbar}{\pi}e^{-\mathcal{A}_t/\hbar} - \frac{i\hbar}{4\pi}e^{-2\mathcal{A}_t/\hbar} + \ldots \tag{4.11}$$

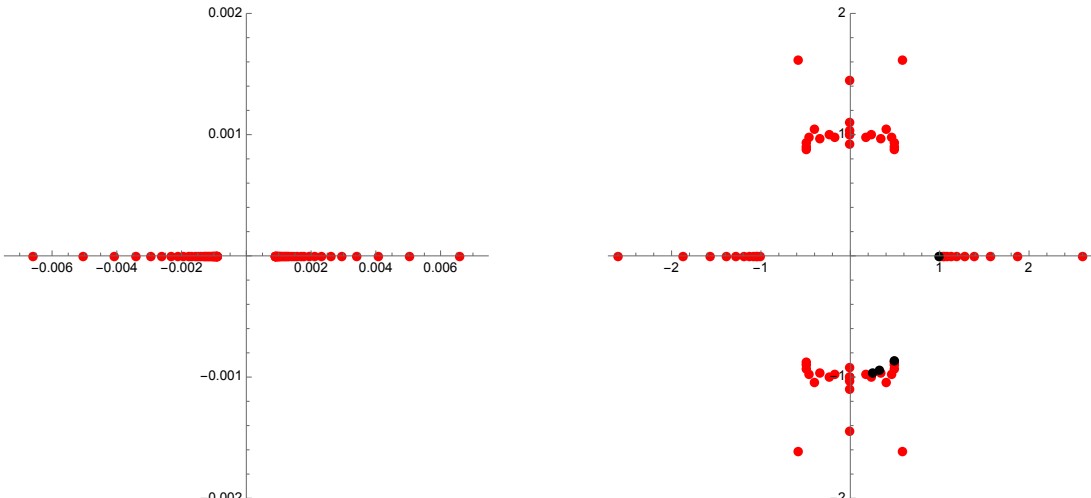

**Figure 4**: The Borel singularities for $\mathcal{F}^{\text{NS}}(\xi;\hbar)$ at $\xi = 1/32 - 10^{-4}$ in the $\xi$-frame. In the left panel we plot the poles of the conventional Padé approximant in the $\zeta$-plane, while in the right panel we plot the poles of the conformal Padé approximant in the $\tau$-plane. In both cases we use the NS free energies up to $n = 145$. The black dots correspond to the location of the singularities $m\mathcal{A}_c$ with $m = 1, 2, 3, 4$ in the $\tau$-plane.

where the lateral resummations are made along the positive imaginary axis. Since

$$s_+ = s_- \mathfrak{S}, \tag{4.12}$$

where the Stokes automorphism is given by (3.137), we deduce that

$$\dot{\Delta}_{m\mathcal{A}_t} \mathcal{F}^{(0)} = \frac{\mathrm{i}\hbar}{\pi} \mathcal{F}_{\mathcal{A}_t}^{(m)}, \quad m = 1, 2. \tag{4.13}$$

This is consistent with (3.120) and gives the following value of Stokes constants

$$\mathsf{S}_{m\mathcal{A}_t}^F(\hbar) = \frac{\mathrm{i}\hbar^2}{\pi}, \tag{4.14}$$

which are indeed independent of $m$ as anticipated before.

To calculate the Stokes coefficients at $\ell\mathcal{A}_c$ of the NS free energies, we can choose $\xi$ very close to $\xi = 1/32$, for instance at $\xi = 1/32 - 10^{-4}$, so that $\pm\mathcal{A}_c$ are the dominant Borel singularities, as in the right panel of Fig. 4. We find then

$$(s_+ - s_-) \mathcal{F}^{(0)} = s_- \left\{ \frac{\mathrm{i}\hbar}{2\pi} \mathrm{e}^{-\mathcal{G}/\hbar} - \frac{\mathrm{i}\hbar}{8\pi} \mathrm{e}^{-2\mathcal{G}/\hbar} - \frac{\hbar}{8\pi^2} \mathsf{D}\mathcal{G}\mathrm{e}^{-2\mathcal{G}/\hbar} + \cdots \right\}, \tag{4.15}$$

up to two instanton orders, where the lateral resummations are made along the positive real axis. We deduce that

$$\dot{\Delta}_{\ell\mathcal{A}_c} \mathcal{F}^{(0)} = \frac{\mathrm{i}\hbar}{2\pi} \mathcal{F}_\ell^{(\ell)}, \qquad \ell = 1, 2. \tag{4.16}$$

To obtain this result we have taken into account that

$$\frac{1}{2!} \dot{\Delta}_{\mathcal{A}_c}^2 \mathcal{F}^{(0)} = -\frac{\hbar}{8\pi^2} \mathsf{D}\mathcal{G}\mathrm{e}^{-2\mathcal{G}/\hbar}. \tag{4.17}$$

A similar numerical calculation can be done for the Wilson loop, and we find, up to two instantons

$$(s_+ - s_-)\omega^{(0)} = s_- \left\{ -\frac{\mathrm{i}\hbar}{2\pi} \mathsf{D}\omega^{(0)}\mathrm{e}^{-\mathcal{G}/\hbar} - \frac{\hbar}{4\pi^2} \left( \frac{\hbar}{2} \mathsf{D}^2\omega^{(0)} - \mathsf{D}\omega^{(0)}\mathsf{D}\mathcal{G} \right) \mathrm{e}^{-2\mathcal{G}/\hbar} \right.$$
$$\left. + \frac{\mathrm{i}\hbar}{4\pi} \mathsf{D}\omega^{(0)}\mathrm{e}^{-2\mathcal{G}/\hbar} + \cdots \right\}. \tag{4.18}$$

We deduce from this discontinuity formula

$$\dot{\Delta}_{\ell\mathcal{A}_c}\omega^{(0)} = \frac{\mathrm{i}\hbar^2}{2\pi}\omega_\ell^{(\ell)}, \qquad \ell = 1, 2, \tag{4.19}$$

where we have taken into account that

$$\frac{1}{2}\dot{\Delta}_{\mathcal{A}_c}^2\omega^{(0)}(\hbar) = -\frac{\hbar}{4\pi^2} \left( \frac{\hbar}{2}\mathsf{D}^2\omega^{(0)} - \mathsf{D}\omega^{(0)}\mathsf{D}\mathcal{G} \right)\mathrm{e}^{-2\mathcal{G}/\hbar}. \tag{4.20}$$

The results (4.16), (4.19) are in complete agreement with the conjectures (3.122) and (3.123), and the following identical Stokes coefficients are found

$$\mathsf{S}_{\ell\mathcal{A}_c}^F(\hbar) = \mathsf{S}_{\ell\mathcal{A}_c}^\omega(\hbar) = \frac{\mathrm{i}\hbar^2}{2\pi}, \tag{4.21}$$

which are also independent of $\ell$ as anticipated before. This value is in fact a consequence of (3.11) (e.g. the power $\hbar^2$ is due to the shift $-2$ in the Gamma function). Since the Delabaere–Pham formula follows from the conjectures (3.122), (3.123) and (3.135), we conclude that this formula holds for the quantum $A$-period of the double well model. This is of course known from the results in [6, 7].

Let us now consider the structure of Borel singularities in the $\xi_D$-frame, i.e. the *dual conifold* frame associated to the period $t_D$. The Borel singularities for $\mathcal{F}^{\mathrm{NS,D}}$ are the same as the ones in the conifold frame, given by (4.6). This is seen in the left panel of Fig. 5, where we plot the singularities of the Padé approximant of the Borel transform of $\mathcal{F}^{\mathrm{NS,D}}$ at $\xi = 1/64$. For the Wilson loops the structure is different: the singularities $\ell\mathcal{A}_c$ on the real axis have disappeared; instead we find singularities at the points $m\mathcal{A}_t$ on the imaginary axis, which is in perfect agreement with (3.121). This is seen for example in the right panel of Fig. 5, where we plot the singularities of the Padé approximant of the Borel transform of $\omega^{\mathrm{D}}$ at $\xi = 1/64$. The singularities at $\pm\mathcal{A}_t$ are clearly visible.

## 4.2 Local $\mathbb{P}^2$

We will now consider the simplest, non-trivial local CY, namely local $\mathbb{P}^2$. Its mirror curve (2.17) can be written as

$$\mathrm{e}^x + \mathrm{e}^p + \mathrm{e}^{-x-p} + \xi = 0, \tag{4.22}$$

and the appropriate parameter describing the moduli space is $z = \xi^{-3}$.

The necessary information for the direct integration of the HAE for local $\mathbb{P}^2$ is collected in the Appendix of the companion paper [29], and will not be repeated here. In the case of the NS free energies, this procedure has been studied in [23, 27, 62]. The integration of the HAE for the Wilson loops in local $\mathbb{P}^2$ has been recently developed in [24]. The only boundary condition that is needed in this case is regularity at the conifold point $z = -1/27$. As an example, one finds,

$$w_1(S, z) = \frac{1}{144z^2(1 + 27z)}(2S - z^2 - 54z^3), \tag{4.23}$$

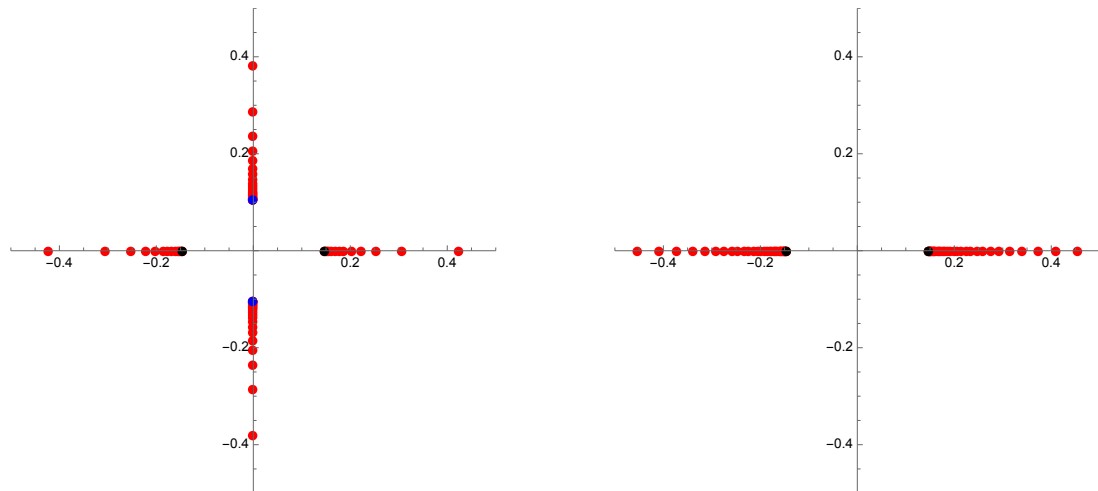

**Figure 5**: The Borel singularities for $\mathcal{F}^{\mathrm{NS}}(\xi;\hbar)$ (left) and $\omega(\xi;\hbar)$ (right) at $\xi = 1/64$ in the $\xi_D$-frame. In both cases perturbative expansions up to $n = 145$ are used. The blue dots in the imaginary axis show the location of $\pm\mathcal{A}_t$. The black dots in the real axis show the location of $\pm\mathcal{A}_c$.

We note that the conventions for the NS free energies that we are using are *different* from the ones used in [27]. In particular, our NS free energy $F_n$ is obtained from the one in [27] by multiplying by $(-1)^n$.

As in [29], in our analysis of the resurgent structure we will focus on the region of moduli space

$$-\frac{1}{27} < z < 0. \tag{4.24}$$

Let us first consider the free energies and Wilson loops in the large radius frame. For values of $z$ close to the conifold point, we find Borel singularities for both $\mathcal{F}^{\mathrm{NS}}$ and $\omega$ at

$$\ell\mathcal{A}_c, \qquad \ell \in \mathbb{Z}, \tag{4.25}$$

where

$$\mathcal{A}_c = \frac{2\pi}{\sqrt{3}}t_c, \tag{4.26}$$

and $t_c$ is the flat coordinate at the conifold. It can be written in a compact way in terms e.g. of a Meijer function,

$$t_c(z) = \frac{3\sqrt{3}}{2\pi}\left(\frac{G^{3,2}_{3,3}\left(-27z \left|\begin{smallmatrix} \frac{1}{3}, \frac{2}{3}, 1 \\ 0, 0, 0 \end{smallmatrix}\right.\right)}{2\sqrt{3}\pi} - \frac{4\pi^2}{9}\right). \tag{4.27}$$

This is the singularity (3.12) expected just from the conifold behavior (in this case one has $\mathfrak{b} = -3$, with the opposite sign to what is used in [29]).

As an example, in Fig. 6 we show the singularities of the Padé approximants of $\mathcal{F}^{\mathrm{NS}}$ and $\omega$ in the Borel plane. The singularities at $\pm\mathcal{A}_c$ in the real axis are clearly visible (higher values of $\ell$ in (4.25) can be unveiled if one uses conformal maps, as in the previous example of the double

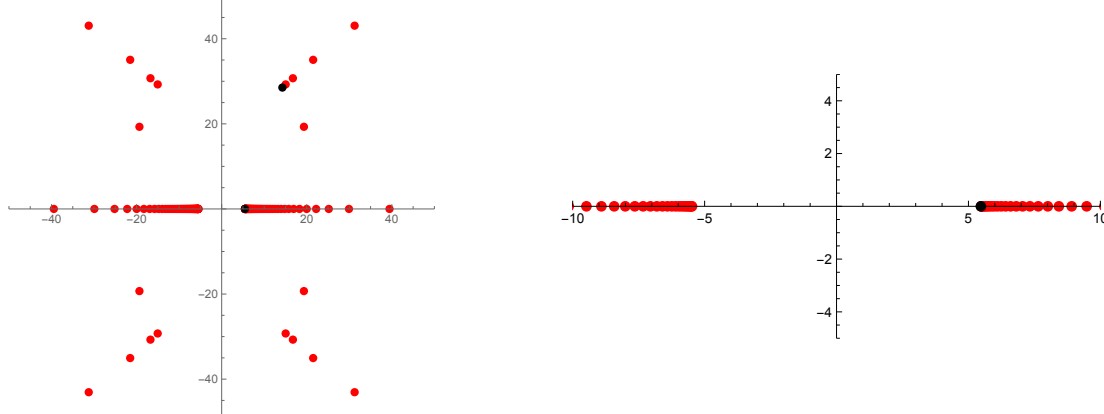

**Figure 6**: The Borel singularities for $\mathcal{F}^{\text{NS}}(z; \hbar)$ (left) and $\omega(z; \hbar)$ (right) for $z = -10^{-2}$, in the large radius frame. The black dots in the real positive axis show the location of $\mathcal{A}_c$. The black dot in the first quadrant of the figure on the left is the point (4.31).

well quantum mechanical model or as illustrated in [29]). In addition, there are singularities of $\mathcal{F}^{\text{NS}}$ at points of the form

$$\pm 2\pi\mathrm{i}t(z) + 4\pi^2 m + \ell \mathcal{A}_c(z), \qquad m, \ell \in \mathbb{Z}, \tag{4.28}$$

where

$$t(z) = -\log(z) + 6z \, {}_4F_3\left(1, 1, \frac{4}{3}, \frac{5}{3}; 2, 2, 2; -27z\right) \tag{4.29}$$

is the classical flat coordinate at large radius. We will write these points as

$$\mathcal{A}_{\ell,n}^{\pm} = \pm 2\pi\mathrm{i}\mathrm{Re}(t(z)) + 4\pi^2\left(n + \frac{1}{2}\right) + \ell \mathcal{A}_c(z), \qquad n, \ell \in \mathbb{Z}. \tag{4.30}$$

For example, for $z = -10^{-2}$ one can see in Fig. 6 a singularity in the first quadrant at

$$\mathcal{A}_{0,-1}^{+} = 2\pi\mathrm{i}\mathrm{Re}(t(z)) + 2\pi^2 - \mathcal{A}_c(z). \tag{4.31}$$

As we approach $z = 0$, the points belonging to the set (4.28) become easier to detect (note however that not all these points are realized as actual singularities). In the case of the Wilson loop, if $z$ is sufficiently close to $z = 0$, new singularities appear in a similar tower, but the value $\ell = 0$ is not allowed, i.e. the singularities are of the form $\mathcal{A}_{\ell,n}^{\pm}$ with $\ell \neq 0$. For example, when $z = -15 \cdot 10^{-6}$, we find for the free energy and in the first quadrant the points $\mathcal{A}$, as well as the singularities

$$\mathcal{A}_{0,0}^{+}, \quad \mathcal{A}_{0,1}^{+}, \quad \mathcal{A}_{1,-1}^{+}, \quad \mathcal{A}_{1,0}^{+}. \tag{4.32}$$

However, for the Wilson loop only the last two singularities, with $\ell \neq 0$, appear.

The determination of the Stokes coefficients associated to these singularities is difficult. However, for the singularities at $\ell \mathcal{A}_c$, which are on the real axis, one can perform a numerical calculation with high precision. In the case of the NS free energies, a numerical calculation up to third instanton order gives

$$\begin{aligned}
(s_+ - s_-)\mathcal{F}^{(0)} = s_-\Bigg\{ &-\frac{\mathrm{i}\hbar}{2\pi}\mathrm{e}^{-\mathcal{G}/\hbar} + \frac{\mathrm{i}\hbar}{8\pi}\mathrm{e}^{-2\mathcal{G}/\hbar} - \frac{\hbar}{8\pi^2}\mathrm{D}\mathcal{G}\mathrm{e}^{-2\mathcal{G}/\hbar} - \frac{\mathrm{i}\hbar}{8\pi^2}\mathrm{D}\mathcal{G}\mathrm{e}^{-3\mathcal{G}/\hbar} \\
&-\frac{\mathrm{i}\hbar^2}{48\pi^3}\left(\mathrm{D}^2\mathcal{G} - \frac{3}{\hbar}(\mathrm{D}\mathcal{G})^2\right)\mathrm{e}^{-3\mathcal{G}/\hbar} - \frac{\mathrm{i}\hbar}{18\pi}\mathrm{e}^{-3\mathcal{G}/\hbar} + \cdots \Bigg\},
\end{aligned} \tag{4.33}$$

where the lateral resummations are made along the positive real axis. We conclude that

$$\dot{\Delta}_{\ell\mathcal{A}_c}\mathcal{F}^{(0)} = -\frac{\mathrm{i}\hbar}{2\pi}\mathcal{F}^{(\ell)}_\ell, \qquad \ell = 1, 2, 3. \tag{4.34}$$

In deducing this result, we have taken into account (4.17) as well as that

$$\frac{1}{3!}\dot{\Delta}^3_{\mathcal{A}_c}\mathcal{F}^{(0)} = -\frac{\mathrm{i}\hbar^2}{48\pi^3}\left(\mathsf{D}^2\mathcal{G} - \frac{3}{\hbar}(\mathsf{D}\mathcal{G})^2\right)\mathrm{e}^{-3\mathcal{G}/\hbar},$$

$$\dot{\Delta}_{\mathcal{A}_c}\dot{\Delta}_{2\mathcal{A}_c}\mathcal{F}^{(0)} = \frac{\hbar}{8\pi^2}\mathsf{D}\mathcal{G}\mathrm{e}^{-3\mathcal{G}/\hbar}. \tag{4.35}$$

A similar numerical calculation can be done for the Wilson loop, and we find, up to two instantons,

$$(s_+ - s_-)\omega^{(0)} = s_-\left\{\frac{\mathrm{i}\hbar}{2\pi}\mathsf{D}\omega^{(0)}\mathrm{e}^{-\mathcal{G}/\hbar} - \frac{\hbar}{4\pi^2}\left(\frac{\hbar}{2}\mathsf{D}^2\omega^{(0)} - \mathsf{D}\omega^{(0)}\mathsf{D}\mathcal{G}\right)\mathrm{e}^{-2\mathcal{G}/\hbar}\right.$$

$$\left. - \frac{\mathrm{i}\hbar}{4\pi}\mathsf{D}\omega^{(0)}\mathrm{e}^{-2\mathcal{G}/\hbar} + \cdots\right\}. \tag{4.36}$$

We deduce from this discontinuity formula

$$\dot{\Delta}_{\ell\mathcal{A}_c}\omega^{(0)} = -\frac{\mathrm{i}\hbar^2}{2\pi}\omega^{(\ell)}_\ell, \qquad \ell = 1, 2. \tag{4.37}$$

The results (4.34), (4.37) are again in agreement with the conjectures (3.122) and (3.123), and the Stokes coefficients are

$$\mathsf{S}^F_{\ell\mathcal{A}_c}(\hbar) = \mathsf{S}^\omega_{\ell\mathcal{A}_c}(\hbar) = -\frac{\mathrm{i}\hbar^2}{2\pi}, \tag{4.38}$$

which are identical and also independent of $\ell$ as anticipated before. This value agrees with the one obtained in (4.21) for the double-well, up to an overall sign, since both follow from the conifold behavior (3.11). From our verification of (3.122), (3.123) and (3.135), we can deduce that the Delabaere–Pham formula holds for the quantum $A$-period of local $\mathbb{P}^2$.

Let us now consider the structure of Borel singularities in the *conifold* frame. The Borel singularities for $\mathcal{F}^{\mathrm{NS}}$ are very similar to the ones in the large radius frame. For the Wilson loops the structure is different: the singularities $\ell\mathcal{A}_c$ on the real axis have disappeared, in agreement with (3.121). We find instead towers of singularities at the points $\mathcal{A}^\pm_{\ell,n}$, as can one see for example in Fig. 8. There we plot the singularities of Padé approximants for $z = -10^{-2}$ (left) and $z = -15 \cdot 10^{-6}$ (right). One can easily see the singularities at $\mathcal{A}^+_{\ell,0}$ with $\ell = -1, \cdots, 2$ and $\mathcal{A}^+_{0,1}$, in the plot on the left, and the singularities $\mathcal{A}^+_{\ell,0}$, $\ell = -1, 0, 1$ and $\mathcal{A}^+_{0,n}$ with $n = 1, 2$, in the plot on the right.

## 5 Conclusions and outlook

In this paper we have studied the resurgent structure of the quantum periods, for general quantum curves with one single modulus. To do this, we have followed the ideas of [25–27] and we have found exact trans-series solutions to the HAE governing the NS free energies and the Wilson loop vevs. These solutions give in a sense the formal skeleton for the resurgent structure of the theory. We have conjectured that there is a special family of multi-instanton solutions which appear as trans-series associated to actual singularities, and leads generically to discontinuities

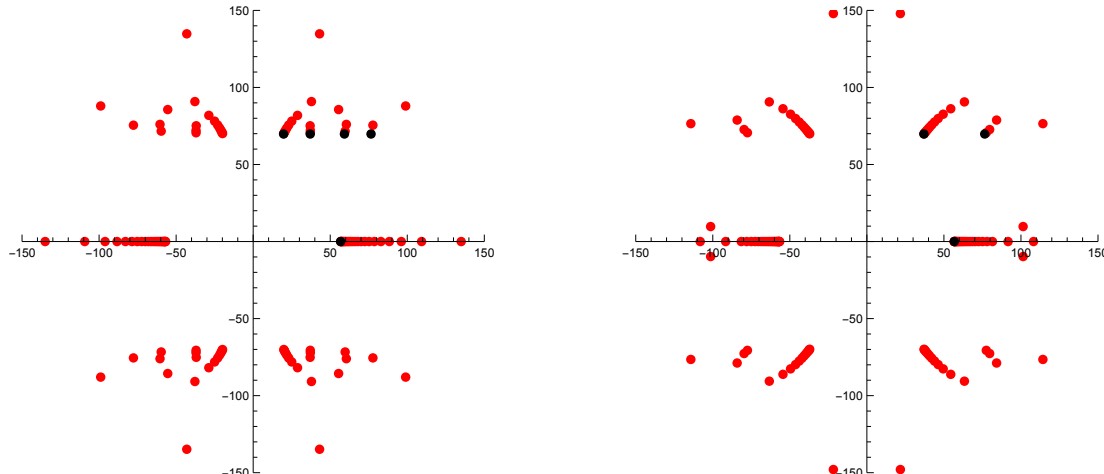

**Figure 7**: The Borel plane of $\mathcal{F}^{\mathrm{NS}}(z;\hbar)$ (left) and $\omega(z;\hbar)$ (right) in the large radius frame, for $z = -15 \cdot 10^{-6}$. The black dots in the first plot show the location of the points $\mathcal{A}_c$, $\mathcal{A}_{0,0}^+$, $\mathcal{A}_{0,1}^+$, $\mathcal{A}_{1,-1}^+$, $\mathcal{A}_{1,0}^+$. In the plot on the right for the Wilson loop, the black dots represent $\mathcal{A}_c$ and the points $\mathcal{A}_{1,-1}^+$, $\mathcal{A}_{1,0}^+$). We used 122 terms in the sequence of $\mathcal{F}_n$ and 132 points in the sequence of $\omega_n$.

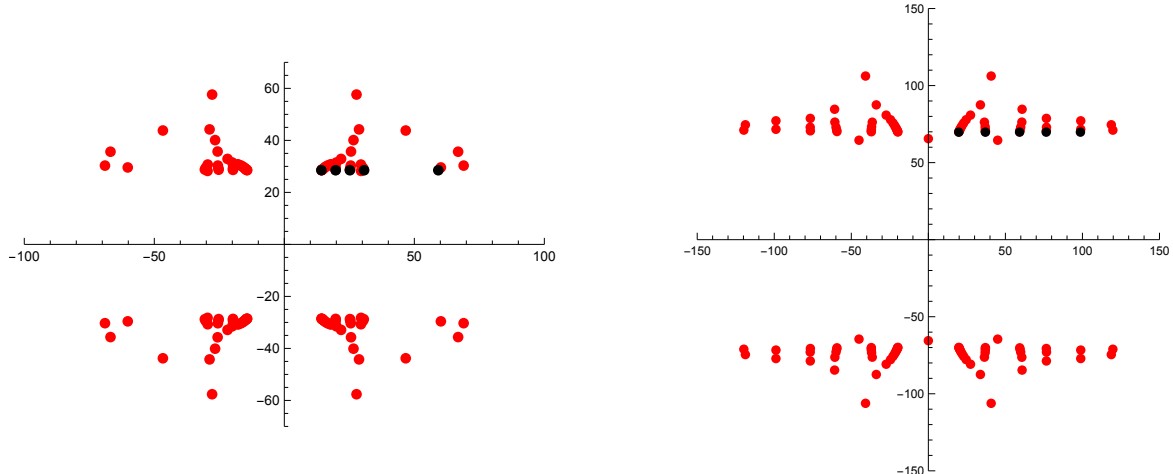

**Figure 8**: The Borel plane of $\omega(z;\hbar)$ in the conifold frame, for $z = -10^{-2}$ (left) and $z = -15 \cdot 10^{-6}$ (right). The black dots in the first plot show the location of the points $\mathcal{A}_{\ell,0}^+$ with $\ell = -1, \cdots, 2$ and $\mathcal{A}_{0,1}^+$, while in the second plot they indicate the points $\mathcal{A}_{1,m}^+$, $m = -1, 0$ and $\mathcal{A}_{0,n}^+$ with $n = 0, 1, 2$. We take 130 terms in the sequence $\omega_n$.

of the Delabaere–Pham form. This conjecture is motivated by the formal structure of the HAE and the behavior at the conifold, and it can be verified in examples. We have given empirical evidence for this conjecture for the quantum periods of the double well in quantum mechanics, and of local $\mathbb{P}^2$.

Our conjecture only describes the formal aspects of the resurgent structure. One has still to determine the actual location of Borel singularities and the Stokes constants associated to them. This information is expected to be related to the structure of the BPS states of a "dual"

supersymmetric theory. The reason is that, in many interesting cases, the complex curve is the Seiberg–Witten curve of a supersymmetric theory, or a mirror curve describing a Calabi–Yau compactification of type II string theory. For a given value of the moduli of the curve, the Borel singularities of the quantum periods should tell us what are the stable BPS states of the supersymmetric theory, at that point in moduli space. The corresponding Stokes constants should be closely related to BPS invariants, counting those BPS states.

In some cases this correspondence between resurgence and BPS states is known to follow from the common mathematical structures underlying the two problems. Let us take for example the case of polynomial potentials for one-dimensional Schrödinger operators. The Borel singularities of the quantum periods correspond to "geodesic cycles" [8], but these are precisely the cycles that support stable BPS states [14, 63]. In the case of quantum Seiberg–Witten curves, the relation between resurgent structures and BPS states was verified in detail in [16] in the case of $SU(2)$ super Yang–Mills theory, and then in [64] for the $SU(2)$ theory with $N_f = 1$. However, as suggested in [15, 16], it should hold more generally. In particular, we expect it to be true in the case of mirror curves, and could shed new light on the difficult problem of identifying and counting BPS states of type IIA compactifications on toric CYs [65]. This problem has been addressed recently with various techniques. On one hand, generalizations of [14] to the local CY case [66] have produced increasingly richer results, see [67, 68] for recent examples. On the other hand, techniques based on attractor flows have been recently applied to local $\mathbb{P}^2$ [69]. It would be very interesting to compare these approaches to the resurgent point of view used in this paper. We could extend the methods developed here to other examples, like local $\mathbb{F}_0$, where detailed comparisons should be possible thanks to the recent results of [67]. Our calculations of Borel singularities and Stokes constants have been so far numerical and of limited reach, and our numerical techniques should be pushed further. Conversely, one could use the results of these other approaches to obtain precise predictions for the resurgent structure of quantum periods.

Another direction to explore is the extension of exact WKB methods to the more general situations considered in this paper. For example, an extension of the Voros–Silverstone connection formula to the difference equations appearing in quantum mirror curves would make it possible to calculate discontinuities from first principles.

## Acknowledgements

We would like to thank Boris Pioline and David Sauzin for useful communications and discussions, and also Alexander van Spaendonck for a careful reading of the manuscript. M.M. would like to thank the Physics Department of the École Normale Supérieure (Paris) for hospitality during the completion of this paper. The work of M.M. has been supported in part by the ERC-SyG project "Recursive and Exact New Quantum Theory" (ReNewQuantum), which received funding from the European Research Council (ERC) under the European Union's Horizon 2020 research and innovation program, grant agreement No. 810573. JG is supported by the Startup Funding no. 3207022203A1 and no. 4060692201/011 of the Southeast University.

## A  Useful formulae for the double well potential

The holomorphic anomaly setting for the double well potential was introduced in [31]. Here we recall some ingredients and we reformulate it in the language we have used so far (in [31] the holomorphic anomaly equations were studied by means of modular forms).

We recall that the complex moduli space will be parametrized in this example by the energy $\xi$. The classical periods solve the Picard–Fuchs equation

$$\left[ 6 + \xi\,(32\xi - 1)\,\frac{\partial^2}{\partial \xi^2} \right] \Pi = 0. \tag{A.1}$$

The solutions can be written in terms of hypergeometric functions,

$$
\begin{aligned}
t =&\, \xi\,_2F_1\left( \frac{1}{4}, \frac{3}{4}, 2, 32\xi \right) = \xi + 3\xi^2 + 35\xi^3 + \frac{1155\xi^4}{2} + \frac{45045\xi^5}{4} + \cdots \\
t_D =&\, 2\sqrt{2}\pi \left( \frac{1}{32} - \xi \right) {}_2F_1\left( \frac{1}{4}, \frac{3}{4}; 2; 1 - 32\xi \right) \\
=&\, \frac{1}{3} + 2\xi \left( \log\left( \frac{\xi}{2} \right) - 1 \right) + \xi^2 \left( 6\log\left( \frac{\xi}{2} \right) + 17 \right) + \cdots .
\end{aligned}
\tag{A.2}
$$

This choice of $A$ period defines what we will call the $\xi$-frame. The prepotential $F_0(t)$ is defined by

$$\frac{\partial F_0}{\partial t} = t_D, \tag{A.3}$$

and it is given by the expansion

$$F_0(t) = \frac{t}{3} + t^2 \left( \log\left( \frac{t}{2} \right) - \frac{3}{2} \right) + \frac{17t^3}{3} + \frac{125t^4}{4} + \frac{3563t^5}{12} + \frac{29183t^6}{8} + \cdots . \tag{A.4}$$

The discriminant is

$$\Delta(\xi) = \xi^2(1 - 32\xi). \tag{A.5}$$

This leads to two singular points, which we will call conifold points. The first one is $\xi = 0$, while the "dual" conifold point is

$$\xi = \frac{1}{32}. \tag{A.6}$$

For this value of $\xi$, the energy of the particle equals the height of the barrier between the two wells. We define a dual energy as

$$\xi_D = \frac{1}{32} - \xi, \tag{A.7}$$

such that $\xi_D = 0$ is the dual conifold point. The classical period $t_D$ can be expanded around $\xi_D = 0$ as

$$t_D = \sqrt{2}\pi \tilde{t}_D, \qquad \tilde{t}_D = 2\xi_D + 6\xi_D^2 + 70\xi_D^3 + \cdots \tag{A.8}$$

Note that $\tilde{t}_D/2$ has the same expansion in terms of $\xi_D$, than $t$ in terms of $\xi$. Physically, the dual situation corresponds to inverting the double-well and exchanging the perturbative and the tunneling cycles, up to an overall factor, and the dual model can be used to describe the inverted quartic oscillator [28, 31]. The frame in which $t_D$ defines the $A$-period will be called the $\xi_D$-frame.

The holomorphic ingredients for this one-modulus special geometry are the following. The propagator $S$ satisfies (2.19), where the Yukawa coupling is

$$C_\xi(\xi) = \frac{2}{\xi(1 - 32\xi)}, \tag{A.9}$$

and the universal functions appearing in (2.19) are

$$\mathfrak{s}(\xi) = \frac{1}{24}(1 - 96\xi), \qquad \mathfrak{f}(\xi) = \frac{1}{576}(1 + 96\xi). \tag{A.10}$$

The holomorphic limit of the propagator in the $\xi$-frame is given by

$$S(\xi) = -\frac{1}{C_\xi(\xi)}\frac{t''(\xi)}{t'(\xi)} + 4\xi - \frac{1}{24}. \tag{A.11}$$

The dual propagator reads

$$S_D(\xi_D) = \frac{1}{C_\xi(\xi_D)}\frac{\tilde{t}''_D(\xi_D)}{\tilde{t}'_D(\xi_D)} + \frac{1}{12} - 4\xi_D \tag{A.12}$$

(the change of sign is due to the fact that $\partial_{\xi_D} = -\partial_\xi$).

The holomorphic anomaly equations (2.23) can be used to solve the NS free energies $F_n$. The first quantum correction (2.22) reads in this case

$$F_1(\xi) = -\frac{1}{24}\log\Delta(\xi). \tag{A.13}$$

and it serves as the initial condition of the HAE. The NS free energies $F_n$ of higher order $n \geq 2$ can be written as

$$F_n(S, \xi) = C_\xi^{2n-2}\sum_{k=0}^{2n-3} S^k p_k^{(n)}(\xi), \tag{A.14}$$

where $p_k^{(n)}(\xi)$ are polynomials in $\xi$. The term with $k = 0$ is the holomorphic ambiguity. It is given by the following ansatz with $3n - 2$ unknowns

$$f_n(\xi) = C_\xi^{2n-2}p_0^{(n)}(\xi) = C_\xi^{2n-2}\sum_{j=0}^{3n-3} c_j\xi^j. \tag{A.15}$$

The holomorphic ambiguities in the NS free energies can be fixed by the behavior at the conifold points [31]. First of all, at the conifold point $\xi = 0$ we impose the standard boundary conditions that the holomorphic limit $\mathcal{F}_n$ behave as (2.26), where $t = t_c$, and $\mathfrak{a} = 2$, $\mathfrak{b} = 1$. On the other hand, the holomorphic limit of the free energies in the "dual" conifold frame $\mathcal{F}_n^D$ are obtained as

$$\mathcal{F}_n^D = F_n\left(\mathcal{S}_D, \frac{1}{32} - \xi_D\right). \tag{A.16}$$

At the dual conifold point $\xi_D = 0$, we impose additional boundary conditions. They are given again by the general expression (2.26), with $t_c = \tilde{t}_D$, $\mathfrak{a} = 1$ and $\mathfrak{b} = -2$. By using the boundary conditions at $\xi = 0$ and at $\xi_D = 0$ we find an overdetermined system for the unknowns appearing in the holomorphic ambiguity (A.15), and all of them can be fixed. One obtains, for example

$$F_2(S, \xi) = \frac{1}{288}\frac{(1 - 48\xi)^2}{\xi^2(1 - 32\xi)^2}S - \frac{\xi^2}{\Delta^2}\left(2\xi^2 - \frac{11}{72}\xi + \frac{79}{34560}\right). \tag{A.17}$$

Similarly, the NS Wilson loops can be solved from the holomorphic anomaly equations (2.34). As in the case of local CY manifolds, we have

$$w_0(\xi) = \log(\xi), \tag{A.18}$$

which gives the initial condition of the HAE. The Wilson loops of higher order $n \geq 1$ can be written as

$$w_n(S, \xi) = C_\xi^{2n} \sum_{k=0}^{2n-1} S^k q_k^{(n)}(\xi), \tag{A.19}$$

where $q_k^{(n)}(\xi)$ are polynomials in $\xi$. The term with $k = 0$ is the holomorphic ambiguity. It is given by the following ansatz with $3n$ unknowns

$$w_n^h(\xi) = C_\xi^{2n} q_0^{(n)}(\xi) = C_\xi^{2n} \sum_{j=0}^{3n-1} c_j \xi^j. \tag{A.20}$$

The boundary conditions at the conifold point $\xi = 0$ are such that the holomorphic limit $\omega_n$ should behave as

$$\omega_n \sim -\frac{1}{2^{2n} n\, \xi^n} + \mathcal{O}(\xi^{1-n}). \tag{A.21}$$

In addition, at the dual conifold point $\xi_D = 0$, the holomorphic limit of the Wilson loops in the "dual" conifold frame

$$\omega_n^D = w_n\left(\mathcal{S}_D, \frac{1}{32} - \xi_D\right) \tag{A.22}$$

should behave as

$$\omega_n^D \sim \mathcal{O}(\xi_D^0). \tag{A.23}$$

The solutions to the HAE (2.34) with the holomorphic ambiguity (A.20) expand at $\xi = 0$ as

$$\omega_n(\xi) = \mathcal{O}(\xi^{-2n}) \tag{A.24}$$

and at $\xi_D = 0$ as

$$\omega_n^D(\xi_D) = \mathcal{O}(\xi_D^{-2n}). \tag{A.25}$$

Therefore the gap conditions (A.21), (A.23) provide $3n + 1$ independent equations which are enough to fix the unknown coefficients. Using these conditions we find, for example,

$$w_1(S, \xi) = -\frac{1 - 48\xi}{12\xi^2(1 - 32\xi)} S - \frac{1}{288\Delta(\xi)}. \tag{A.26}$$

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
