# Peer review of "On the resurgent structure of quantum periods"

_SciPost Physics_

## Round 3 · Referee Report · Anonymous (Referee 1) · 2023-4-12

Strengths

  1. The computations are thorough.
  2. The paper is clearly written.

Weaknesses

  1. Very technical
  2. Some features of the results are obscured.

Report

Non-perturbative completions to physical observables are notoriously both hard to compute and of key importance in understanding global, e.g. monodromy, properties for physical observables.

In this work the authors consider the resurgence structure of quantum periods, which can be thought of as the quantization of classical periods associated with integrals of usual differentials on curves. Two specific cases are considered here: quantum periods associated with WKB quantum mechanics and the quantum mirror curve associated with $\mathbb{P}^2$.
This is a companion paper to 2211.01403, by the same authors, where more emphasis is put on the topological string side of the story and more details are provided.

A unifying picture is obtained from studying the holomorphic anomaly equation (HAE) satisfied by the NS free energy and the Wilson loop vevs (sort of ''generating functions'' for quantum periods).
Once this system of equations is set into place, the authors are able to construct a multi-instanton transseries solution for both the NS free energy and the Wilson loop.

By computing the so-called Alien derivative on their conjectured transseries coefficients, the authors are able to reproduce the Delabaere-Pham formula showing how general this result is, following essentially from the resurgence structure of multi-instantons.

Finally, the authors provide many convincing evidences that their conjectured transseries expansion is indeed correct by numerically studying the cases of the double well potential in QM and the local $\mathbb{P}^2$.

The paper is clearly written and most certainly deserves publication although I wonder whether it would be more suitable for a different journal, for example Annales Henri Poincar\'e.

Requested changes

  1. I found it rather difficult to pin down the main key equations/results of this paper. It would be perhaps helpful for the reader to present already at the level of the introduction what the key results are, e.g. the multi-instanton transseries expansion for the NS free energy and Wilson loop and the general Delabaere-Pham formula.

  2. Secondly, I am afraid section 3 was rather hard to follow. It is a very technical yet necessary section, since it contains all the meat for the calculation, however, I wonder whether it would be best to relegate some of the most technical points to an appendix.

  3. Finally, it might be useful to remind the reader unfamiliar with the Stokes automorphism, what this object actually does, i.e. it connects across two different Stokes wedges.

  • validity: -
  • significance: -
  • originality: -
  • clarity: -
  • formatting: -
  • grammar: -

Author:  Jie Gu  on 2023-04-25  [id 3614]

(in reply to Report 1 on 2023-04-12)
Category:
remark

  1. We will box the important formulas and refer to these formulas in the introduction.
  2. After boxing the results we hope that this is clearer and more readable.
  3. We will add some explanation in the beginning of section 3.3.

---

## Round 3 · Referee Report · Anonymous (Referee 2) · 2023-4-20

Strengths

  1. General closed-form (exact) results for the multi-instanton sectors of the trans-series of the NS free energies and Wilson loop vevs. This provides a much more complete picture of their non-perturbative completion, as compared to previous work.

  2. The authors present the derivations of their computations in a lot of detail. In general they specify when a particular result is conjectured and not derived.

  3. The resurgent analysis of the multi-instanton trans-series provides a direct study of the Delabaere-Pham formula, which is then then thoroughly confirmed in a quantum mechanical and a quantum mirror curve example.

Weaknesses

  1. Notation is not consistent throughout, specially after introducing the negative action instantons. The technical nature of the paper makes it difficult to follow if different notation is used for the same object.

  2. In parts of the work, specially in section 3, it is difficult to separate the main results from their computations/derivations.

  3. It is sometimes difficult to understand when the authors are focusing on particular cases (such as specific boundary conditions) or general cases.

Report

This paper presents a thorough study of the resurgent properties of quantum periods associated to quantum curves of genus one. This is achieved via the analysis of the holomorphic anomaly equations (HAEs), which can be used to obtain the NS free energies and the Wilson loop vevs. These are obtained as asymptotic expansions, then upgraded to trans-series, which define implicitly the quantum periods. Section 2 presents a brief introduction to quantum curves and the HAEs, which are the main building blocks to be used in the computations of Section 3. It also presents a brief summary of the work. Section 3 then includes a very detailed analysis of the non-perturbative completion of the NS free energies and Wilson loop vevs, with closed form for general instanton number thoroughly analysed and derived. The resurgent structure of these quantities is then analysed/conjectured. Finally a trans-series and its resurgent properties is obtained for the quantum periods. Section 4 is devoted to the study of two examples, the quantum mechanical double well, and the mirror curve of local \mathbb{P}^{2}, which corroborates the results obtained in the previous sections. The paper ends with a summary of the work and some open questions.

The paper is generally well written and clear, well referenced and very detailed, and without any major weaknesses. Although this work comes as a companion paper to another work from the same authors, it is my opinion that this work stand perfectly well on its own, and includes novel analysis of the quantum periods, which will certainly be useful for different quantum problems/different curves.

It is my opinion that this paper meets the expectations and generally all the acceptance criteria, and it should be published in SciPost after some minor revision. In order to be published I believe the authors need to review the consistent use of their notation, and perhaps separate their main results from the corresponding derivation, as at the moment their main results are buried sometimes in the middle of the computations and are difficult to find.

Requested changes

There are somechanges that I would request the authors take into consideration. I also have some questions for the authors, which can be found below.

  1. t appears in equation (2.20) but it is not defined until later. There should be some mention of what it is here.

  2. there is a typo on the last line of page 5: I believe that "w.r.t" should not be there.

  3. Keeping track of the different types of S used (propagator, its holomorphic limit and Stokes coefficients) is sometimes difficult.

  4. On page 12, first line after eq (3.25), what is D_S?

  5. In the computations of pages 13-17, y is treated as an independent variable which is then taken to be Sigma(C). Is this correct? The variable y is not clearly explained in this proof, and the authors do not explicitly state its final value in some of the derivations.

  6. On the second line of page 19 I think the wrong equation is being referenced.

  7. On the third line of page 20, for which n,m is the condition valid?

  8. The superscript in (3.105) and (3.106) seems to have reverted back to the notation prior to introducing the negative action instantons, but this is not clear. I suggest the authors review the notation they use throughout to be consistent, and if they do need to revert back to previously used notation, they should state it explicitly.

  9. Is the condition on eq (3.108) still true if one includes both types of instantons (and not only the (\ell|0) ones)?

  10. Below equation (3.109) the authors mention using (3.27), but haven't they used either (3.92) or (3.42)?

  11. When generally calling an equation that is on a previous page, to improve readability quite a lot it would be very useful to state in very few words what that equation is.

  12. Is the resurgent analysis in section 3.3 for the perturbative series alone (meaning that it related the perturbative series with non-perturbative sectors, but the relation between non-perturbative sectors is not given)?

  13. What happens at alpha=0, does the perturbative series truncate or is it still asymptotic?

  14. Would it be possible to include more information on where (3.124) comes from?

  15. The results (3.122) and (3.124) are different from the alien derivatives expected for a two parameter trans-series, why is this the case?

  16. The resurgent analysis in pages 23 and 24 seems to be related to the resurgence of an implicit function (see Delabaere's paper on this subject), is this the case? Can these results be written with that in mind?

  17. It is often not clear when the choice (3.104) was taken and when then results are more general. I suggest the authors review this and explicitly state the generality of the results.

  • validity: high
  • significance: high
  • originality: high
  • clarity: good
  • formatting: excellent
  • grammar: excellent

Author:  Jie Gu  on 2023-04-25  [id 3613]

(in reply to Report 2 on 2023-04-20)
Category:
remark
answer to question
correction
pointer to related literature

  1. You are right. We will add explanation below (2.20).
  2. You are right. We will remove it.
  3. Sorry for the confusion. The three different S's are distinguished by Roman, calligraphic, and sans serif fonts.
  4. This is a typo. D_S should be \partial_S instead.
  5. That is correct. In many places we only need coefficients of powers of y, so the final value of y is irrelevant. We will make it more explicit in the text. For instance below (3.48) there is a typo, and [y^n]f(y) should be the coefficient of the term y^n instead of the term itself.
  6. You are right. It should be referred to (3.82).
  7. It is true for any integer values of n,m.
  8. We will make this more explicit. For instance we will add the following explanation below (3.104): "Note that here and in the following, whenever we are in the pure instanton sector we will revert to the notation $F^{(n)}$, $w^{(n)}$ to reduce the clutter."
  9. We conjecture it still holds when all types of instantons, including in the mixed sectors, are included.
  10. In fact, both (3.42) and (3.27) are needed. More precisely, we have used (3.42) and also taken into account (3.27).
  11. Good suggestion. We shall check.
  12. We did resurgent analysis for both perturbative and non-perturbative sectors. The resurgent structure is completely encoded in the alien derivatives. We conjecture in section 3.3 the alien derivatives of perturbative series in (3.122), (3.124), which are later checked numerically in section 4. The alien derivatives of non-perturbative series are given in (3.126), (3.127). In fact, as the non-perturbative series can be expressed in terms of perturbative series, the alien derivatives of the former can in principle be derived from the alien derivatives of the latter, which we checked in many cases but did not write down the details in the paper.
  13. \alpha is a parameter of the instanton action, as shown in (3.3), i.e. it is part of the instanton sector data. So the perturbative series is not affected by \alpha. Perhaps you refer to non-perturbative series? Then yes, at \alpha=0, the non-perturbative series truncates to the form in (3.14).
  14. We will explain in the updated text that (3.124) can be derived from (3.122) with the map $\hbar\rightarrow -\hbar$ after taking into account that $\mathcal{F}^{(\ell)}{\ell}(-\hbar) = \mathcal{F}^{(0|\ell)}(\hbar)$ and $w^{(\ell)}{\ell}(-\hbar) = w^{(0|\ell)}(\hbar)$. There is a typo though in (3.124). There should be a minus sign on the right hand side in the first line.
  15. Indeed it is, since e.g. (2.31) can be regarded as the definition of an implicit resurgent function. We are using however a slightly more general result which is the chain rule for alien derivatives (3.132). As in the case of standard derivatives, this chain rule allows one to calculate the derivative of the implicit function, which is what we do in (3.133)-(3.134). We will add an additional reference to the book by Candelpergher et al, where these issues are discussed.
  16. You probably have in mind the alien derivatives for a two-parameter trans-series solution of a second order ODE. In our view, there is no reason why a general two-parameter trans-series should satisfy those particular relations, since there are many sources for such trans-series. In our case, our trans-series does not satisfy an ODE w.r.t. the variable in which we do the expansion. It satisfies a PDE w.r.t. additional parameters S and z, so we are doing "parametric resurgence". One can still derive some sort of bridge equation in that case, as we mention in our paper "Exact multi-instantons...". For this reason, the alien derivatives we obtain are not that dissimilar from the ones appearing in the ODE case, but they are certainly not identical.
  17. Whenever the choice (3.104) is taken a subscript $\ell$ is attached. Nevertheless we will try to make this more transparent in the text.

---

## Editorial Decision

resubmitted